# A tripartite organelle platform links growth factor receptor signaling to mitochondrial metabolism

Deborah Mesa[1,2,12], Elisa Barbieri[2,12], Andrea Raimondi [3,4,12], Stefano Freddi [1,2], Giorgia Miloro[2], Gorana Jendrisek[1,2], Giusi Caldieri[2], Micaela Quarto [1,2], Irene Schiano Lomoriello[1,2], Maria Grazia Malabarba[1,2], Arianna Bresci [5], Francesco Manetti [5], Federico Vernuccio [5], Hind Abdo[6], Giorgio Scita [1,6], Letizia Lanzetti [7,8], Dario Polli [5,9], Carlo Tacchetti[3,10], Paolo Pinton [11], Massimo Bonora [11], Pier Paolo Di Fiore [1,2,13] ✉ & Sara Sigismund [1,2,13] ✉

One open question in the biology of growth factor receptors is how a quantitative input (i.e., ligand concentration) is decoded by the cell to produce specific response(s). Here, we show that an EGFR endocytic mechanism, non-clathrin endocytosis (NCE), which is activated only at high ligand concentrations and targets receptor to degradation, requires a tripartite organelle platform involving the plasma membrane (PM), endoplasmic reticulum (ER) and mitochondria. At these contact sites, EGFR-dependent, ER-generated $Ca^{2+}$ oscillations are sensed by mitochondria, leading to increased metabolism and ATP production. Locally released ATP is required for cortical actin remodeling and EGFR-NCE vesicle fission. The same biochemical circuitry is also needed for an effector function of EGFR, i.e., collective motility. The multiorganelle signaling platform herein described mediates direct communication between EGFR signaling and mitochondrial metabolism, and is predicted to have a broad impact on cell physiology as it is activated by another growth factor receptor, HGFR/MET.

Epidermal growth factor (EGF) receptor (EGFR) signaling controls multiple cellular phenotypes, including proliferation, migration, differentiation, apoptosis, and stem cell regulation[1,2]. This variety of outputs is, in part, achieved through the multiplicity of ligands and/or EGFR heterodimerization with other ErbB family receptors[2]. However, the same ligand can induce multiple cellular responses not only in different cellular contexts, but also in the same cell type as a function

of its concentration and duration of exposure[3,4]. How the cell decodes the quantitative growth factor input into a specific cellular response is not completely understood.

We recently discovered a close crosstalk between EGFR endocytosis and interorganelle communication[5]. Clathrin-mediated endocytosis (CME) is the preferred route at limiting EGF concentrations (≤1 ng/ml), while at higher EGF concentrations (≥10 ng/ml) non-

[1]Department of Oncology and Hematology-Oncology, Università degli Studi di Milano, Milan, Italy. [2]IEO, European Institute of Oncology IRCCS, Milan, Italy. [3]Experimental Imaging Centre, IRCCS San Raffaele Hospital Scientific Institute, Milan, Italy. [4]Università della Svizzera italiana (USI), Faculty of Biomedical Sciences, Institute for Research in Biomedicine, Bellinzona, Switzerland. [5]Department of Physics, Politecnico di Milano, Milan, Italy. [6]IFOM, The AIRC Institute of Molecular Oncology, Milan, Italy. [7]Department of Oncology, University of Torino Medical School, Candiolo, Turin, Italy. [8]Candiolo Cancer Institute, FPO-IRCCS, Candiolo, Turin, Italy. [9]CNR Institute for Photonics and Nanotechnology (CNR-IFN), Milan, Italy. [10]Vita-Salute San Raffaele University, Milan, Italy. [11]Department of Medical Sciences, Section of Experimental Medicine and Laboratory for Technologies of Advanced Therapies (LTTA), University of Ferrara, Ferrara, Italy. [12]These authors contributed equally: Deborah Mesa, Elisa Barbieri, Andrea Raimondi. [13]These authors jointly supervised this work: Pier Paolo Di Fiore, Sara Sigismund. ✉e-mail: pierpaolo.difiore@ieo.it; sara.sigismund@ieo.it

clathrin endocytosis (NCE) is also activated[6,7]. Since CME sustains signaling and the EGF-induced mitogenic response by recycling receptors back to the plasma membrane (PM) and NCE attenuates signaling by targeting receptor to lysosomal degradation, the relative contribution of each route to EGFR internalization dictates signaling intensity and thus cellular response[8].

EGFR-NCE involves interorganelle communication by way of contact sites between the PM and endoplasmic reticulon (ER) that are dependent on the cortical ER-resident protein Reticulon-3 (RTN3)[5]. While the long isoform of RTN3 is implicated in endosome maturation and autophagy[9,10], the short isoform is involved in EGFR-NCE[5]. The ER-PM contacts are formed upon EGFR activation and serve as anchors for PM tubular invaginations (TIs) and hot spots of localized calcium signaling triggered by $Ca^{2+}$ released from the ER via the inositol-3-phosphate receptor (IP3R). This $Ca^{2+}$ response is essential for TI fission and subsequent release of NCE vesicles into the cytosol[5]. However, given the pleiotropic roles of $Ca^{2+}$ it is possible that the EGFR-NCE $Ca^{2+}$ response has wider implications in the cell.

Herein, we provide evidence that EGFR-NCE internalizing structures participate in PM-ER-mitochondria contact sites, which constitute a multiorganelle signaling platform where $Ca^{2+}$ oscillations are sensed by mitochondria, leading to an upregulation of mitochondrial cofactors (NAD(P)H and FAD) and ATP production. We took advantage of different tools to measure mitochondrial activity, both globally (i.e., with dyes detecting mitochondrial membrane potential and label-free two-photon excitation fluorescence (TPEF) detecting endogenous nonlinear fluorescence of mitochondrial cofactors) and in a spatially resolved manner (i.e., exploiting calcium probes and the ATP-dependent enzyme, luciferase, targeted to specific cellular compartments). The increase in mitochondrial energetics, in turn, regulates cortical actin remodeling, which is needed for the completion of EGFR-NCE internalization and for inducing collective cell migration. This work directly links EGFR signaling to mitochondrial metabolism via the establishment of a physical and functional interaction between organelles, and demonstrates how a quantitative signal detected at the PM can be deconvoluted into specific cellular responses. This mechanism is relevant for different growth factor receptors, namely EGFR and hepatocyte growth factor receptor (HGFR or MET), and in different epithelial cell contexts, and it is thus predicted to have a broad impact on cellular behavior.

## Results

### Tripartite ER-PM-mitochondria contact sites are regulated by high-dose EGF

CD147 is a specific NCE cargo that co-internalizes with EGFR in the presence of high-dose EGF[5]. Immuno-electron microscopy (EM) of HeLa cells revealed that CD147-NCE internalizing structures are in close contact with ER stacks which, in turn, are in contact with mitochondria (Fig. 1a and Supplementary Fig. 1a). 3D tomographic reconstruction confirmed the topological nature of these tripartite contacts showing that an ER tubule is always interposed between NCE PM invaginations and neighboring mitochondria (Fig. 1b and Supplementary Movie 1 and 2). Quantification of the ER-mitochondrial contacts by EM tomographic reconstruction (Fig. 1c) revealed that high-dose EGF treatment increases the fraction of mitochondria with extended areas of contact with the ER (Fig. 1d), without significantly affecting mitochondrial morphology (Supplementary Fig. 1b). These EGF-induced ER-mitochondrial contacts were located close to the PM (within 1 μm), while contacts further from the PM were unchanged by EGF treatment (Fig. 1e).

To determine the impact of different perturbations on these contact sites, we developed an alternative method for quantifying them that could be applied to multiple samples. This method measures the colocalization of ER and mitochondrial markers in deconvolved fluorescence images of ultra-thin sections (100 nm) of resin-embedded cells (Fig. 1f, left and center, yellow dots). By doing this, while the x-y resolution remains at the diffraction limit (200 nm), an improvement of ~10-fold is obtained in the axial resolution compared vs. conventional confocal microscopy[11]. The accuracy of this approach was confirmed by correlative light electron microscopy (CLEM; Fig. 1f, right), which established a good correlation between the number of contacts identified by fluorescence and by EM (Supplementary Fig. 1c). The fluorescence method confirmed the EM tomography data (Fig. 1e) indicating that only high dose EGF increases the number of ER-mitochondrial contacts near to the PM (Fig. 1g). Furthermore, the knockdown (KD) of the critical EGFR-NCE regulator, RTN3, inhibited the formation of PM proximal ER-mitochondrial contacts, showing that they are dependent on EGFR-NCE (Fig. 1g and Supplementary Fig. 1d). In contrast, the KD of RTN4, a reticulon family member not involved in NCE[5], did not have an inhibitory effect, but instead appeared to slightly increase the contacts near to the PM. This observation is in line with the reported effects of RTN4 KD on general ER morphology[5,12], which are however unrelated to NCE[5].

Thus, EGF appears to induce the formation/stabilization of ER-mitochondria contact sites near to NCE PM internalizing structures. Notably, these tripartite structures are triggered by EGF before the TIs are formed (Fig. 1a). While the temporal sequence does not imply causality, we note that we have previously shown that the abrogation of contact sites by RTN3 KD affects TIs formation[5]. Thus, the formation of the tripartite platform might act upstream of the membrane invagination event.

### High EGF induces PM-restricted $Ca^{2+}$ waves through the buffering activity of mitochondria

We previously showed that high EGF induces $Ca^{2+}$ release from the ER, via the IP3R channel, at NCE PM-ER contact sites using the $Ca^{2+}$ sensor Aequorin[5]. Here, we investigated the contribution of mitochondria to this $Ca^{2+}$ response, using the $Ca^{2+}$ probe, GCaMP6f[13], which at variance with Aequorin permits measurements of $Ca^{2+}$ oscillations at the single-cell level. Using HeLa cells stably transfected with GCaMP6f targeted to the PM inner leaflet (PM-GCaMP6f), we showed that high but not low EGF elicited repetitive $Ca^{2+}$ spikes at the PM (Fig. 2a, b, Supplementary Fig. 2a and Supplementary Movie 3), which were NCE specific, as shown by their dependency on RTN3 and IP3R, but not on RTN4 (Fig. 2b and Supplementary Fig. 2b). Also, the main CME adapter protein, AP2, showed a minor impact on EGF-induced $Ca^{2+}$ oscillations at the PM (Supplementary Fig. 2c, d).

The EGF-induced $Ca^{2+}$ oscillations resemble those observed upon agonist stimulation[14], whose generation and propagation require mitochondrial $Ca^{2+}$ buffering at the ER interface[15–17]. Indeed, inhibition of the mitochondrial calcium uniporter (MCU) complex by siRNA or the selective inhibitor MCUi11[18], abolished the EGF-induced $Ca^{2+}$ waves at the PM, similarly to RTN3 and IP3R KD (Fig. 2c, d and Supplementary Fig. 2e). Importantly, the inhibition of MCU did not reduce the $Ca^{2+}$ signaling response at the PM induced by another receptor agonist, i.e., histamine, a well-studied signaling pathway leading to $Ca^{2+}$ release from the ER[19], arguing for specificity in the EGF-induced response at the PM (Supplementary Fig. 2f), at least in the systems herein tested.

Finally, using the GCaMP6m sensor targeted to the mitochondrial matrix (Mito-GCaMP6m, Fig. 2a right), we demonstrated that high, but not low, EGF induces $Ca^{2+}$ oscillations inside the mitochondria that were dependent on MCU, IP3R and RTN3, but not on RTN4 (Fig. 2e, f and Supplementary Movie 4). Similar results were obtained using mitochondrial-targeted Aequorin (mito-Aequorin, Supplementary Fig. 2g,h)[20]. Thus, mitochondria play a pivotal role in EGF-dependent $Ca^{2+}$ buffering at PM-ER-mitochondria tripartite contact sites, and are required to achieve a productive $Ca^{2+}$ signaling oscillatory response at the PM induced by EGF.

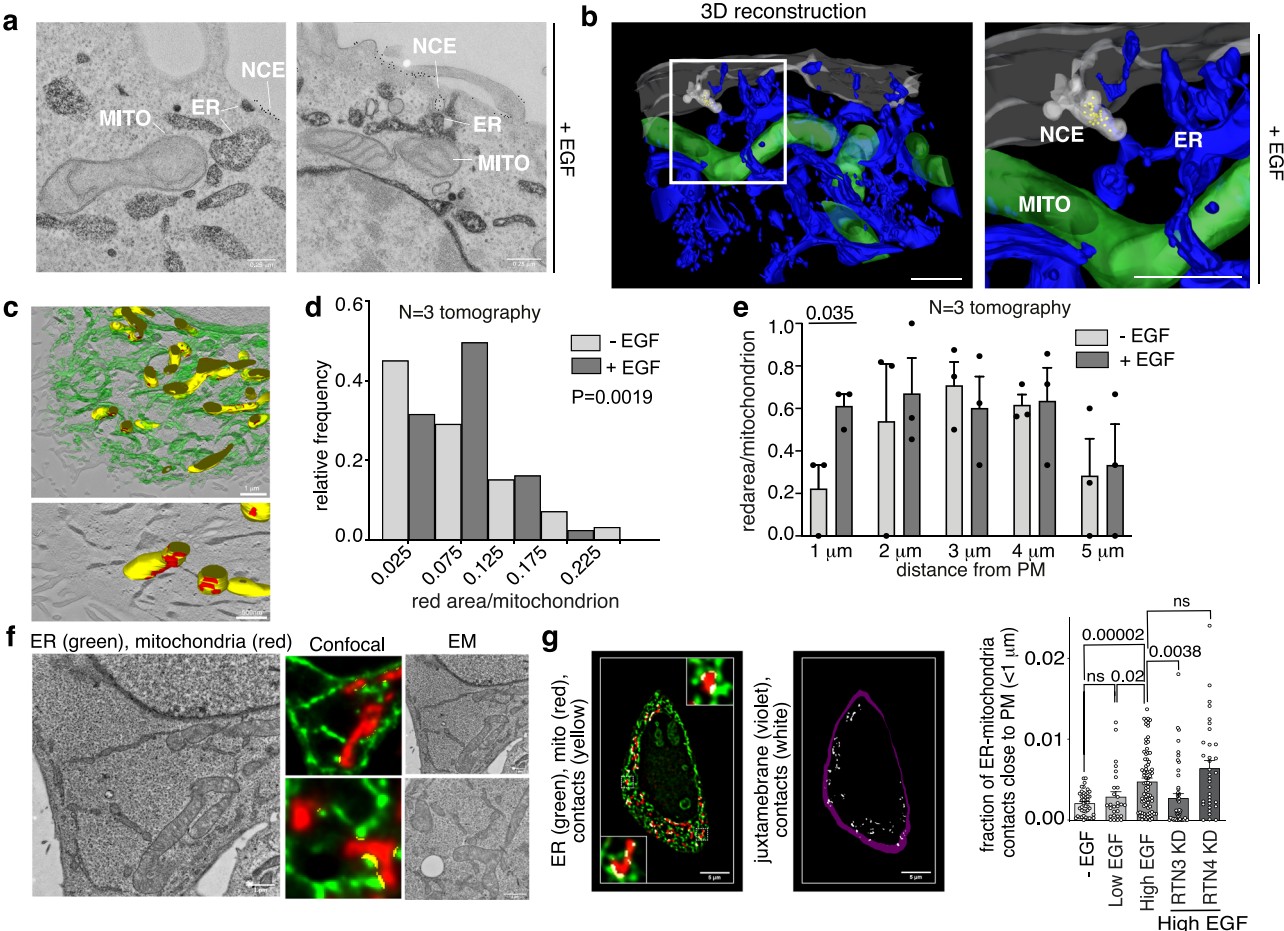

**Fig. 1 | EGFR-NCE involves tripartite PM-ER-mitochondria contact sites.**
**a** Representative immuno-EM image of a tripartite contact between an NCE structure positive for CD147 (gold-immunolabelled), ER (stained with HRP-KDEL) and mitochondrion ($n = 3$). HeLa cells were stimulated with high dose EGF (5 min). Bar, 0.25 μM. **b** Tomographic 3D reconstruction of the cortical region of a HeLa cell as in (**a**) showing the proximity between NCE structures (light gray) containing the cargo CD147 (yellow dots), the ER (blue) and mitochondria (green). Right, magnification. Bar, 1 μm. **c** Upper panel, tomographic 3D reconstruction of the ER (green) and mitochondrial networks (yellow) as in **a**. Lower panel, the ER signal was removed and areas of contact (<20 nm) with mitochondria are highlighted in red. Bars, 1 μM (upper), 500 nm (lower). **d** Quantification of the red area/mitochondrion (>0.05) as in (c) in unstimulated vs. high dose EGF-stimulated cells. Results are expressed as the frequency of mitochondria displaying an ad hoc defined range of red area/mitochondrion ratios relative to the total of mitochondria. P-value, $\chi^2$ test of the absolute values, two-tailed ($N = 3$). **e** The fraction of mitochondria in contact with

the ER, determined as in **c**, is reported as a function of distance from the PM. Mean ± SD. **f** Left, representative deconvolved image of an ultrathin-section of resin-embedded HRP-KDEL-transfected HeLa cells. Center/right, cells stained with FITC-Tyramide (ER in green) and mitotracker (in red). Representative CLEM images showing fluorescence (center) and EM (right) images of the same cell sections ($n = 2$). Contact sites were recognized by ImageJ JACoP-plugin (yellow). Bars, 1 μm. **g** Control, RTN3-KD and RTN4-KD HeLa cells, transfected with HRP-KDEL, were stimulated as indicated. Left, representative deconvolved image of resin-thin-section of HeLa control cells stained as in **f**. Center, ER-mitochondria contacts (white), juxtamembrane area (violet). Right, ER-mitochondria contacts in the jux-tamembrane area quantified with ImageJ JACoP-plugin. Bar, 5 μm. Mean density of ER-mitochondrial contacts within 1 μm from the PM ± SEM is reported. $N$ = number of cells: -EGF/N = 44, LowEGF/N = 28, HighEGF/N = 78, RTN3-KD/N = 50, RTN4-KD/ N = 34. Exact $p$-values are shown (Each Pair Student's $t$ test, two-taileda); ns not significant, n biological replicate. Source data are provided as a Source Data file.

## EGF-dependent Ca²⁺ oscillations induce changes in mitochondrial energetics

EGF-induced Ca²⁺ oscillations inside the mitochondrion are predicted to affect mitochondrial metabolic functions, first and foremost ATP production. Indeed, the accumulation of Ca²⁺ in the mitochondrial matrix stimulates the activity of Ca²⁺-dependent dehydrogenases of the Krebs cycle, resulting in elevation of the mitochondrial membrane potential (ΔΨm) and ATP production[21]. To investigate the potential crosstalk between EGFR signaling and mitochondria, we measured the effect of EGF stimulation on ΔΨm using the potentiometric dye TMRM[22]; since ΔΨm is required for ATP production, changes in this parameter act as a proxy for variations in ATP production. High, but not low, EGF rapidly increased TMRM fluorescence intensity (Fig. 3a, b and Supplementary Movie 5), indicative of an increase in ΔΨm and mitochondrial ATP production. This

response appeared to be dependent on the Ca²⁺ signaling at NCE contact sites since it was inhibited by RTN3, IP3R, or MCU KD and by acute treatment with the MCU inhibitor, MCUi11, but was not decreased by RTN4 or AP2 KD (Fig. 3c and Supplementary Fig. 3a, b).

To confirm the impact of EGF on mitochondrial metabolism, we employed an approach based on label-free TPEF that has been recently characterized as a method to monitor subtle metabolic changes[23,24]. In particular, the TPEF signal scales linearly with the concentration of mitochondrial cofactors, NAD(P)H and FAD, in the irradiated cell volume. Importantly, we observed that high-dose EGF induces a broader distribution of the TPEF signal expanding into the peripheral cytoplasmic areas, whereas the signal in non-stimulated cells and RTN3 KD cells concentrates in the perinuclear region (Fig. 3d, left). Quantification of the TPEF signal confirmed a significantly higher percentage

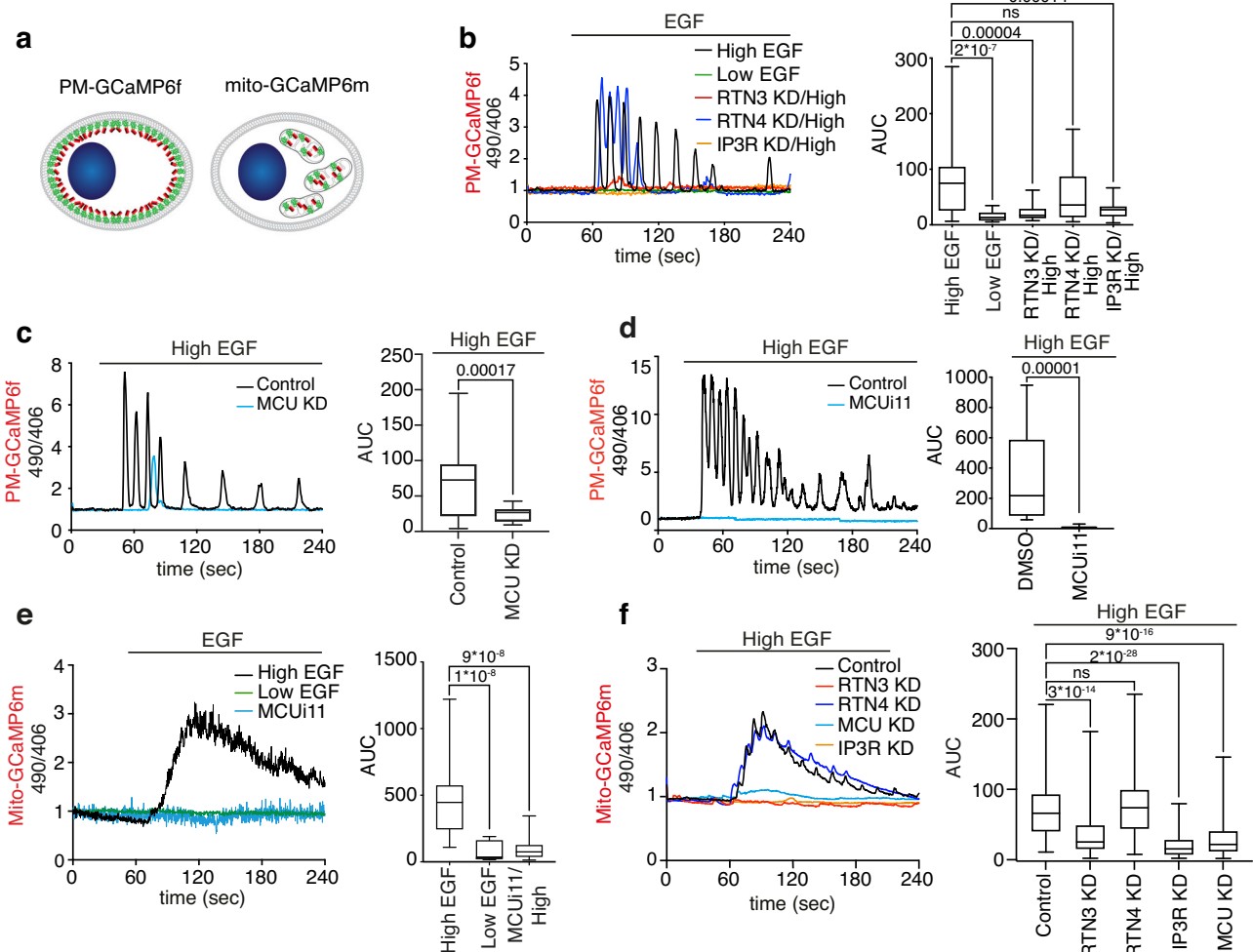

**Fig. 2 | EGFR-NCE induces Ca²⁺ oscillations at the PM which rely on mitochondrial Ca²⁺ buffering activity.** a Schematic of the GCaMP6f Ca²⁺ probe targeted to the PM inner leaflet (PM-GCaMP6f, left) or to inside mitochondria (mito-GCaMP6m, right). b HeLa PM-GCaMP6f cells were stimulated with low dose EGF/N = 31 or high dose EGF in the presence of RTN3-KD/N = 25, RTN4-KD/N = 26 or IP3R-KD/N = 25 or mock control/N = 40 and fluorescence monitored (n = 3). N number of cells. Results (here and in panels c-f) are presented as the ratio of the emission at 490/406 nm. Left, representative single-cell response curves. Right, area under the curve (AUC) is shown. c, d HeLa PM-GCaMP6f cells subjected or not to MCU KD (c) or MCUi11 treatment (d) and stimulated with high dose EGF. c Control/N = 44, MCU-KD/N = 21

(n = 4); (d) Control/N = 20, MCUi11-treated/N = 20, one representative experiment of n = 2 is shown. e HeLa mito-GCaMP6m cells treated with low EGF/N = 12 or high EGF alone/N = 27 or with MCUi11/N = 30 (n = 1). f HeLa mito-GCaMP6m cells subjected or not to the indicated KDs and stimulated with high dose EGF. Control/N = 123; RTN3-KD/N = 135; RTN4-KD/N = 95; IP3R-KD/N = 123; MCU-KD/N = 102 (n = 4). In the box plots, the lower and upper boundaries of the box are the first and third quartiles, with the median annotated with a line inside the box. The whiskers extend to the maximum and minimum values. Exact p-values are shown in all panels (Each Pair Student's t test, two-tailed); ns not significant, n biological replicates. Source data are provided as a Source Data file.

of cell area featuring fluorescent mitochondrial co-enzymes in high dose EGF stimulated cells compared with RTN KD and unstimulated cells (Fig. 3d, right). These results are in line with the idea that high-dose EGF induces an increase in the metabolic activity of peripheral mitochondria by triggering EGFR-NCE.

To investigate further how EGF affects intracellular ATP, we used the ATP-dependent enzyme luciferase, which in the presence of its substrate luciferin emits luminescence as a function of ATP levels[25]. When luciferase was expressed in the cytosol, high-dose EGF resulted in a reduction in luminescence indicating the consumption of cytosolic ATP, which was not observed at low EGF doses (Supplementary Fig. 3c, d). However, in cells expressing luciferase targeted to the PM inner leaflet, a peak of luminescence at the subplasmalemmal level was observed immediately after stimulation with high, but not low, EGF (Fig. 3e, f and Supplementary Fig. 3e), indicative of a local increase in ATP levels. This increase in ATP appears to be dependent on NCE-associated Ca²⁺ signaling, mitochondrial buffering activity and mitochondrial ATP production since it occurred only in the presence of

high-dose EGF and was inhibited by RTN3 KD and inhibitors of IP3R (xestospongin C, XeC), MCU (MCUi-11), mitochondrial ATP synthase (oligomycin, OMY) and mitochondrial ADP/ATP translocase which mediates ATP export from mitochondria (bongkrekic acid, BKA), but not by RTN4 KD or AP2 KD (Fig. 3g, h and Supplementary Fig. 3f, g). Of note, RTN4 KD caused an increase in subplasmalemmal ATP levels that might be explained by the increase in mitochondria-ER contacts sites observed in this condition (Fig. 1g).

The link between mitochondrial metabolism and NCE - but not CME - was corroborated by observations in a HeLa clone (OSLO) which activate only CME and not NCE upon EGF stimulation[5,7]. HeLa OSLO cells stimulated with high dose EGF did not show PM-localized Ca²⁺ oscillations nor changes in ΔΨm and subplasmalemmal ATP levels (Supplementary Fig. 3h–j).

Together, these data suggest that the active EGFR, through NCE and the generation of Ca²⁺ waves, can signal to mitochondria in proximity of NCE sites to increase their production of ATP which is then locally released near the PM.

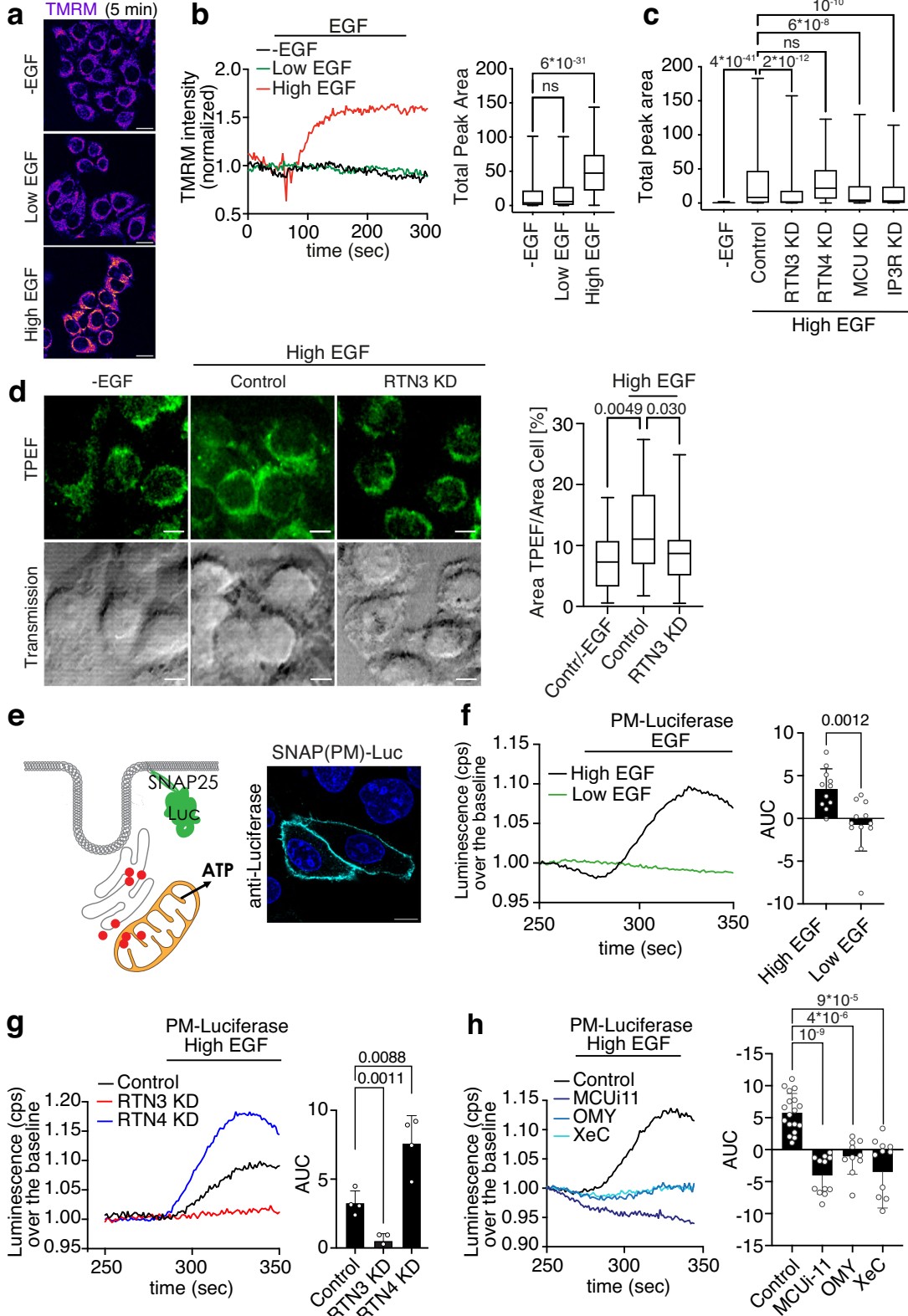

## The crosstalk between NCE and mitochondria is observed also for the HGFR

To understand whether the NCE mechanism is relevant to other growth factor receptors, we investigated HGFR (MET) which is is highly expressed in epithelial cells including HeLa[26]. HGFR is activated by its only known ligand, HGF, and like EGFR it can be ubiquitinated by the E3 ligase Cbl, endocytosed and targeted to lysosomal degradation[27,28]. We

therefore stimulated HeLa cells with saturating doses of HGF and followed CD147 internalization (in vivo) and HGFR localization (staining it with specific antibody after cell fixation). In unstimulated cells, the HGFR signal was mainly localized at the PM, while only a small amount of CD147 signal was detectable in intracellular vesicles (i.e., after acid wash treatment, Fig. 4a). Upon HGF stimulation, HGFR internalization was induced as expected. HGF stimulation also induced the

**Fig. 3 | NCE-associated Ca²⁺ oscillations induce mitochondrial ATP production.**
**a** Representative images of TMRM fluorescence in HeLa cells stimulated or not with low or high dose EGF after 45 s since recording started. Bar, 10 μM. **b** Left, time course of TMRM fluorescence intensity in cells treated as in "a". Right, Total peak area±SD: -EGF/N = 219; LowEGF/N = 217; HighEGF/N = 205; $N$ = number of cells ($n$ = 5). **c** TMRM fluorescence measured in HeLa cells, subjected to the indicated KD or mock control, and stimulated with high dose EGF (5 min). Mock unstimulated cells (-EGF), negative control. Total peak area is reported. N=number of cells: -EGF/N = 68, Control/N = 421, RTN3-KD/N = 316, RTN4-KD/N = 324, IP3R-KD/N = 265, IP3R-KD/N = 228 ($n ≥ 3$). **d** Left, representative TPEF images of HeLa cells mock transfected/unstimulated (Control/-EGF) or subjected to RTN3 KD or mock transfection (Control) and stimulated with high dose EGF (5 min). Images have a dimension of 70 × 70 μm². Scale bar: 10 μm. Right, quantification of the distribution of TPEF signal in HeLa cells treated as in left panel: N, Field of View, -EGF/N = 26, Control/High EGF/N = 32, RTN3-KD/High EGF/N = 27 ($n$ = 3). Statistical significance

was calculated using the non-parametric U-Mann Whitney test, two-tailed. **e** Left, schematic showing luciferase targeted to the PM inner leaflet by fusion with the PM-targeting domain of SNAP25. Right, representative IF staining showing the localization of the SNAP25-luciferase fusion protein [SNAP(PM)-Luc] ($n$ = 3). Bar, 10 μm. **f** Luminescence was measured in HeLa cells expressing PM-Luc stimulated with low or high EGF. Left, representative curves of luminescence over the baseline; cps, count per second. Right, mean AUC ± SD. N=number of coverslips (for panels **f**–**h**): HighEGF/N = 11, LowEGF/N = 12 ($n$ = 2). **g, h** Luminescence was measured in HeLa cells expressing PM-Luc were subjected or not to the indicated KDs (**g**) or inhibitor treatments (**h**) and stimulated with high dose EGF. Results are presented as in **f**. **g** Control/N = 4, RTN3-KD/N = 4, RTN4-KD/N = 4; a representative experiment of $n$ = 4 is shown. (h) Control/N = 18, MCUi11/N = 12, oligomycin (OMY)/N = 10, xestospongin (XeC)/N = 12 ($n ≥ 3$). Box plots (panels **b**–**d**) are defined as in Fig. 2f. Each Pair Student's $t$ test, two-tailed (panels **b, c, f**–**h**); exact $p$-values are shown; ns not significant, n biological replicates. Source data are provided as a Source Data file.

internalization of CD147, which colocalized with intracellular HGFR (Fig. 4a). Immuno-EM revealed that CD147-NCE internalizing PM forms tripartite contacts sites with ER cortical tubules and mitochondria (Fig. 4b). As in the case of EGF, tripartite contact sites are established upon HGF before the invagination is formed (Fig. 4b). Importantly, HGFR endocytosis was inhibited by both RTN3 KD and clathrin KD alone, and further decreased upon the double KD, as quantified by an increase in the ratio of PM to total HGFR (Fig. 4c). These results suggest that HGFR, similarly to EGFR, can be internalized by both CME and NCE mechanisms. Notably, CD147 internalization was monitored and shown to be independent of clathrin, but requiring RTN3, as expected for this NCE specific marker (Fig. 4d). Together, these data show that high dose HGF stimulates the internalization of HGFR through the NCE mechanism in HeLa cells.

Importantly, HGF stimulation induces Ca²⁺ oscillations at the PM, similarly to EGF stimulation, and these oscillations are dependent on NCE (RTN3), the IP3R calcium channel on the ER and mitochondrial buffering activity (MCU) (Fig. 4e). Moreover, these PM-localized Ca²⁺ oscillations induced by HGF led to rapid changes in mitochondrial energetics dependent on the same molecular circuitry (Fig. 4f). These results extend the relevance of NCE beyond the EGFR system, showing that HGFR activates the same circuitry influencing calcium signaling and mitochondrial energetics similarly to EGFR.

## Mitochondrial Ca²⁺ buffering activity and localized mitochondrial ATP production are required for NCE at multiple steps

To investigate the role of the crosstalk between EGFR and mitochondria in NCE, we inhibited the different steps of this signaling and assessed the effects on the internalization of the NCE cargo CD147[5]. Inhibition of Ca²⁺ release (IP3R KD) or mitochondrial buffering activity (MCU KD or MCUi11 inhibitor treatment) inhibited CD147-NCE (Fig. 5a and Supplementary Fig. 4a), without affecting the internalization of the CME cargo transferrin (Tf) (Supplementary Fig. 4b). Next, we acutely inhibited mitochondrial ATP production by short treatment (5 min) with the mitochondrial ATP synthase inhibitor OMY. This treatment inhibited CD147-NCE (Fig. 5b top) as well as EGFR-NCE, but not EGFR-CME. Indeed, OMY inhibited Alexa647-EGF internalization in clathrin KD cells (CME inhibited, NCE active), but not in RTN3 KD cells (CME active, NCE inhibited), showing that it is inhibiting EGFR-NCE and not EGFR-CME (Fig. 5b bottom and Supplementary Fig. 4c, d). In addition, acute OMY treatment did not alter the activation of the EGFR kinase nor its downstream signaling (Supplementary Fig. 4e), and did not induce changes in total cellular ATP levels, at variance with long-term treatments with OMY or Rotenone (Supplementary Fig. 4f), showing that acute treatment with OMY is not altering the overall cytosolic pool of ATP.

Since we showed that high-dose EGF induces a localized increase in ATP at the PM, we investigated the effect of selectively depleting this ATP pool. To this end, we used SNAP25-luciferase, not as an ATP sensor, but as an ATP-consuming enzyme. In the presence of an excess of

luciferin, CD147/EGF internalization was selectively inhibited in cells expressing SNAP25-luciferase, but not in control cells (Fig. 5c and Supplementary Fig. 5a). In contrast, Tf endocytosis remained unaffected (Supplementary Fig. 5b), arguing that NCE selectively requires the PM-localized ATP pool for its execution, at variance with CME.

To understand the role of Ca²⁺ waves and PM-localized ATP in NCE, we studied the formation of NCE TIs upon inhibition of the ER-mitochondrial Ca²⁺-ATP axis. Our previous data indicated that the inhibition of IP3-mediated Ca²⁺ release from the ER obtained by treatment with XeC did not impact on ER-PM contacts of on TI formation[5]. On the contrary, the inhibition of mitochondrial ATP production (short-term OMY), while not affecting ER-PM contact formation, reduced the number of TIs at the PM (Supplementary Fig. 5c, d). These data suggest that mitochondrial ATP is required for PM invagination. Notably, all treatments, either affecting Ca²⁺ release from the ER (XeC), or mitochondrial Ca²⁺ uptake (MCU KD), or mitochondrial ATP production (short-term OMY), resulted in the elongation of NCE TIs to a similar extent as dynamin KD (the GTPase involved in NCE fission, Fig. 5d and Supplementary Fig. 5e)[5]. In contrast, clathrin-coated pits (CCPs) were elongated only upon dynamin KD, which is known to participate in CCP fission during CME (Fig. 5e)[29]. These results argue for a role of Ca²⁺ and ATP in the fission of NCE TIs, while dynamin is involved in both CME and NCE.

To understand if Ca²⁺ and ATP are both required or if Ca²⁺ is needed solely to stimulate mitochondrial ATP production, we performed a CD147 rescue experiment in MCU KD cells where both Ca²⁺ waves and ATP production are inhibited. In these cells, we induced a Ca²⁺ burst independent of the EGF-NCE circuitry by stimulation with histamine (Supplementary Fig. 2e) and/or induced mitochondrial ATP production independently of Ca²⁺ by feeding the Krebs Cycle with succinate (Supplementary Fig. 5f). We found that only the combination of histamine and succinate was able to rescue CD147 internalization in MCU KD cells (Fig. 5f), indicating that Ca²⁺ signaling and mitochondrial ATP act in concert in the execution of EGFR-NCE.

## Ca²⁺ signaling and increased mitochondrial ATP production are required for cortical actin remodeling at NCE sites

Actin polymerization relies on protein machinery that is dependent on Ca²⁺ and ATP[30–32]. While CME requires actin polymerization only when PM tension is elevated[33], different clathrin-independent endocytosis (CIE) mechanisms depend on actin cytoskeleton dynamics for PM deformation and/or for fission[34]. We, therefore, investigated whether the requirement for the Ca²⁺/ATP axis in EGFR-NCE is to sustain cortical actin remodeling. Inhibition of the key regulators of actin polymerization, Arp2/3 (by the CK666 inhibitor) and N-WASP (by KD), impaired CD147/EGF internalization in HeLa cells stimulated with high EGF but did not affect Tf-CME (Fig. 6a and Supplementary Fig. 6a, b). Moreover, Arp2/3 inhibition impaired NCE TI fission (as measured by

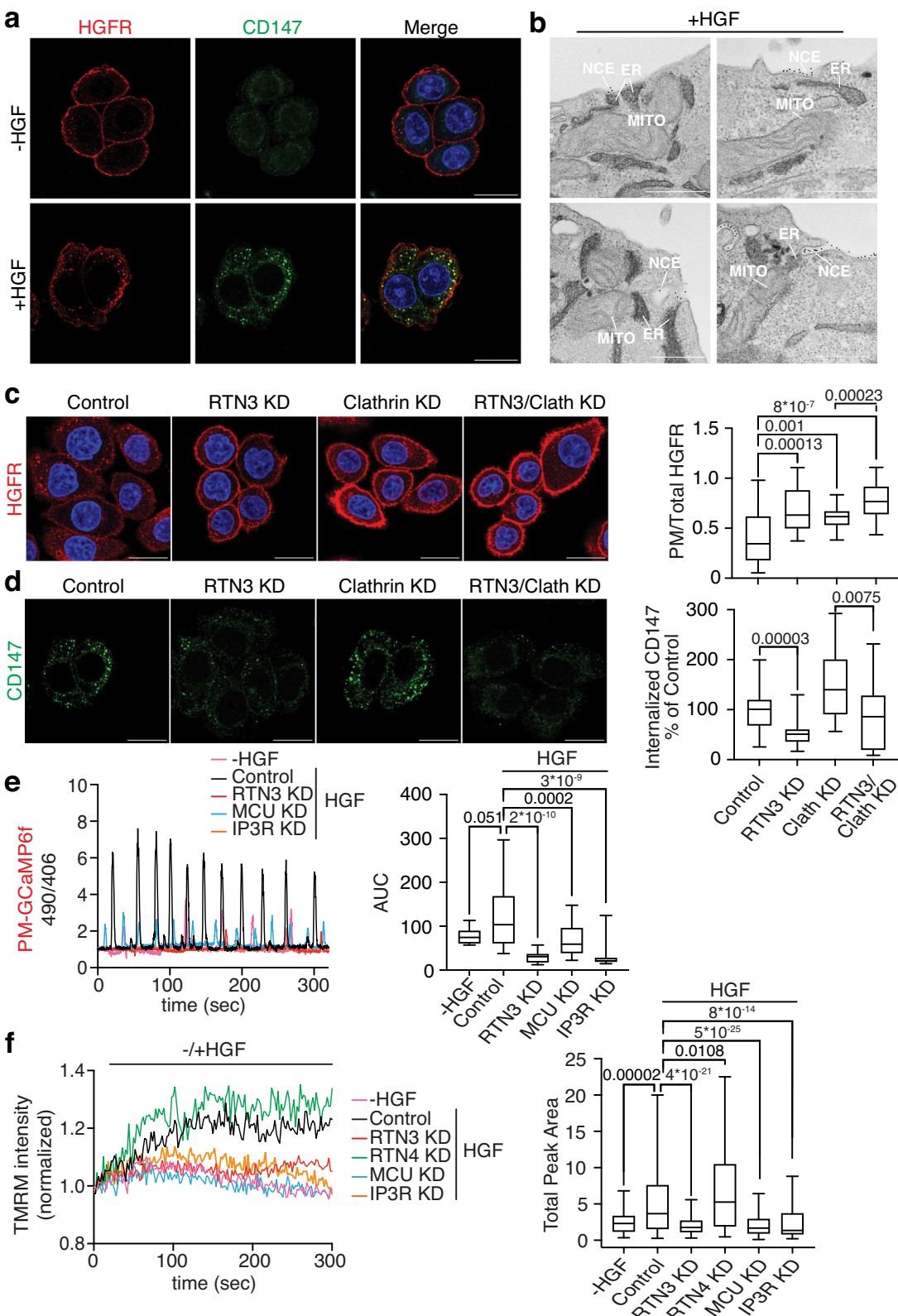

the ratio between long and short TIs), while it had no effect on CCP fission (Fig. 6b, c). Inhibition of actin polymerization was synergistic with dynamin KD in increasing the length of NCE TIs but not of CCPs (Fig. 6b, c), indicating that actin and dynamin cooperate in NCE TI fission.

EGF stimulation is known to increase F-actin remodeling inducing extensive cellular ruffling (Supplementary Fig. 6c, left and middle panels)[35]. Mechanistically, high EGF induces Arp2/3 relocalization to the PM, at variance with low EGF (Supplementary Fig. 6c, right panel). Using a cortical actin polymerization probe called MPAct[36], we found that high, but not low dose of EGF induced extensive cortical actin polymerization which peaks after 5 min of EGF stimulation (Fig. 6d, e and Supplementary Fig. 6d, e). This polymerization was dependent on NCE and the $Ca^{2+}$/ATP axis, as witnessed by its abrogation upon KD

**Fig. 4 | HGFR is internalized via NCE and relies on the same Ca²⁺/ATP circuitry as EGFR. a** CD147 internalization (green) was monitored in vivo in HeLa cells stimulated with HGF (100 ng/ml, 8 min). Control cells were left unstimulated. Acid wash treatment was applied prior to fixation. HGFR was stained with a specific antibody after fixation and permeabilization (red). Blue, DAPI. Bar, 20 μm. **b** Representative immuno-EM image of a tripartite contact between a NCE structure positive for CD147 (immunolabelled with gold), ER (stained with HRP-KDEL) and mitochondrion. HeLa cells were stimulated with HGF (100 ng/ml, 5 min). Images in **a**, **b** are representative of $n = 2$. **c** HGFR internalization was measured in HeLa cells stimulated with HGF (100 ng/ml, 8 min) under different KD conditions. After fixation, HGFR was stained prior to permeabilization to detect PM-HGFR. Cells were then further fixed and permeabilized to enable total HGFR staining. Left, representative IF images of total HGFR (red). Bar, 20 μm. Right, quantification of the ratio PM-HGFR/total-HGFR. Image analysis was performed using ImageJ software. *N*, number of cells, Control/N = 114, RTN3-KD/N = 130, Clathrin-KD/N = 126, RTN3/Clathrin-KD/N = 122 ($n = 2$). **d** CD147 internalization followed as in **a** was quantified

under different KD conditions. Mean integrated fluorescence intensity is expressed as a percentage of control. *N*, number of cells: Control/N = 94, RTN3-KD/N = 105, Clathrin-KD/N = 111, RTN3/Clathrin-KD/N = 94 ($n = 2$). **e** HeLa PM-GCaMP6f cells were subjected to the indicated KDs or a mock control, and stimulated or not with HGF (250 ng/ml). Fluorescence was monitored; results are presented as the ratio of emission at 490/406 nm. Left, representative single-cell response curves. Right, area under the curve (AUC). -HGF/N = 10, Control/N = 37, RTN3-KD/N = 32, MCU-KD/N = 30, IP3R-KD/N = 30 ($n = 1$). **f** TMRM fluorescence was measured in HeLa cells subjected to the indicated KDs or a mock control and stimulated with HGF (100 ng/ml, 5 min) after a 45-s recording period. Mock/unstimulated cells (-HGF), negative control. Left, representative traces. Right, total peak area. -HGF/N = 55, Control/N = 193, RTN3-KD/N = 219, RTN4-KD/N = 148, MCU-KD/N = 276, IP3R-KD/N = 199 ($n = 1$). Box plots (panels c-f) are defined as in Fig. 2f. Exact *p*-values (Each Pair Student's *t* test, two-tailed) are shown; ns not significant, n biological replicates. Source data are provided as a Source Data file.

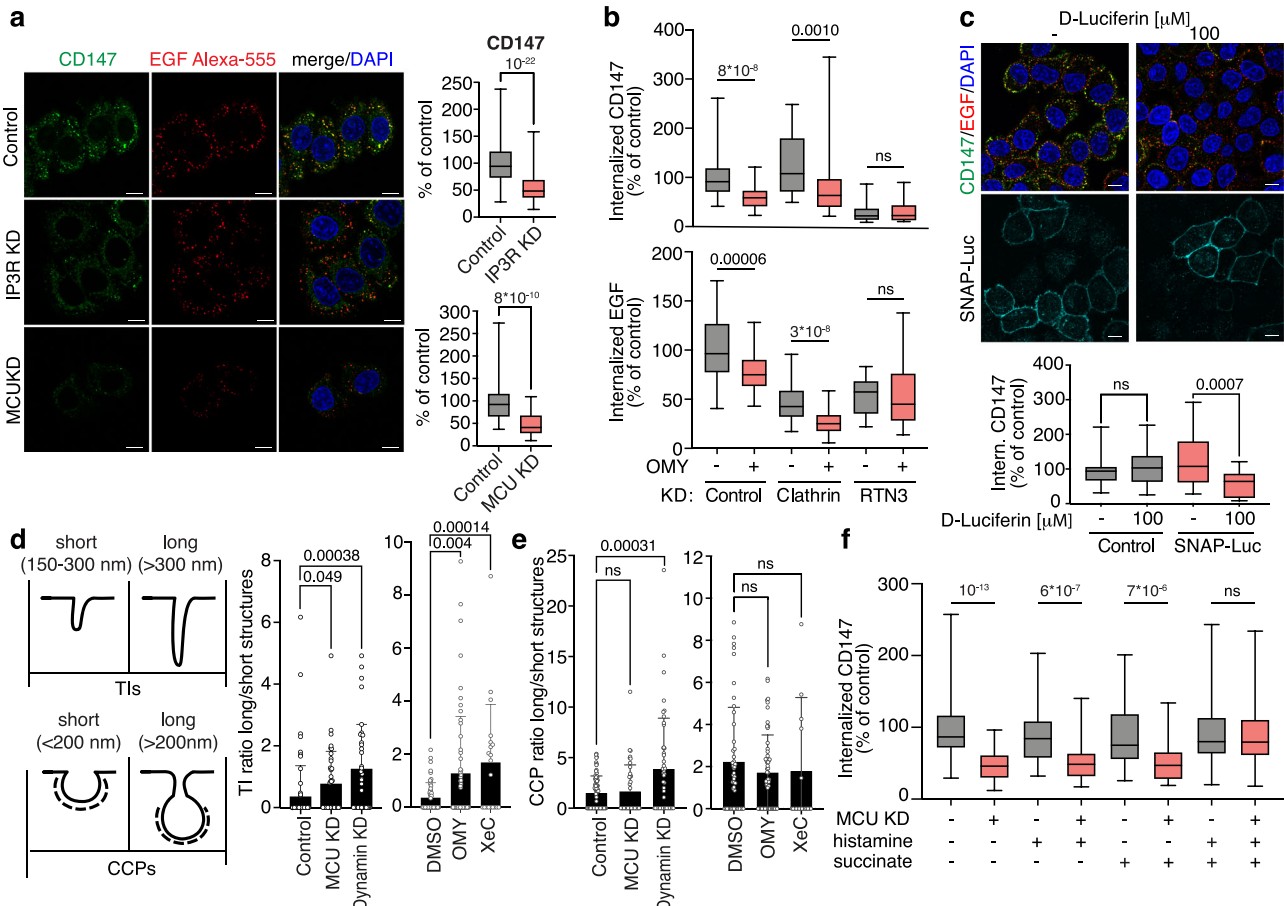

**Fig. 5 | EGF-induced Ca²⁺ waves and mitochondrial ATP production are required for NCE execution. a** CD147 internalization was monitored in vivo by IF in HeLa cells subjected to the indicated KDs or mock control, stimulated with high dose Alexa-555-EGF. Cells were subjected to acid wash prior to fixation to remove membrane-bound antibodies. Left, representative IF images. Blue, DAPI. Bar, 10 μm. Right, quantification of CD147 fluorescence intensity expressed as % of control. *N* = number of cells: Control/N = 699, IP3R-KD/N = 673 ($n = 8$); Control/N = 258, MCU-KD/N = 239 ($n = 5$). **b** Internalization of CD147 and Alexa-555 EGF was monitored as in **a** in HeLa cells subjected to the indicated KDs, treated or not with OMY (1 μM). CD147 and Alexa-555-EGF fluorescence intensity is expressed as % of control. *N* = number of cells: Control/N = 222, Control+OMY/N = 217, Clathrin-KD/N = 201, ClathrinKD+OMY/N = 252 ($n = 4$); RTN3-KD/N = 106, RTN3-KD + OMY/N = 132 ($n = 2$). **c** HeLa cells transfected with PM-Luc, were treated or not with luciferin (100 μM). CD147 internalization was monitored as in **a**. Top, representative IF images, blue, DAPI. Bar, 10 μm. Bottom, CD147 fluorescence intensity expressed as % of control. *N* = number of cells: Control/N = 189, Control+Luciferin/N = 142 ($n = 3$); SNAP-Luc/N = 116, SNAP-Luc+Luciferin/N = 78 ($n = 4$). **d**, **e** EM morphometric

analysis of the length of EGFR gold-positive TIs or CCPs in HeLa cells subjected to the indicated KDs or drug treatments. **d** Left, criteria used to assign TIs or CCPs as short or long. Right, quantification of the ratio of long vs. short NCE-TIs normalized to 100 μm length PM profiles, mean ± SD. N = cell profiles: Control/N = 70, MCU-KD/N = 39, Dyn-KD/N = 35, DMSO/N = 50, OMY/N = 56, XeC/N = 20 ($n ≥ 2$). **e** Quantification of the ratio of long vs. short CCPs normalized to 100 μm length PM profiles. N = cell profiles: Control/N = 70/n = 3; MCU-KD/N = 39/n = 2; Dyn-KD/N = 45/n = 3; DMSO/N = 46/n = 3; OMY/N = 56/n = 3; XeC/N = 20/n = 2. **f** HeLa cells were subjected to MCU KD or mock control and treated or not with histamine and/or succinate. CD147 internalization was monitored as in **a**. Quantification of CD147 fluorescence intensity expressed as % of control. N = number of cells: Control/N = 281, MCU-KD/N = 345, Histamine/N = 177, MCU-KD+Histamine/N = 264, Succinate/N = 169, MCU-KD+Succinate/N = 202, Histamine/Succinate/N = 225, MCU-KD+Histamine/Succinate/N = 230 ($n = 3$). Box plots (panels **a–c**, **f**) are defined as in Fig. 2f. Exact *p*-values (Each Pair Student's *t* test, two-tailed) are shown; ns not significant, n biological replicates. Source data are provided as a Source Data file.

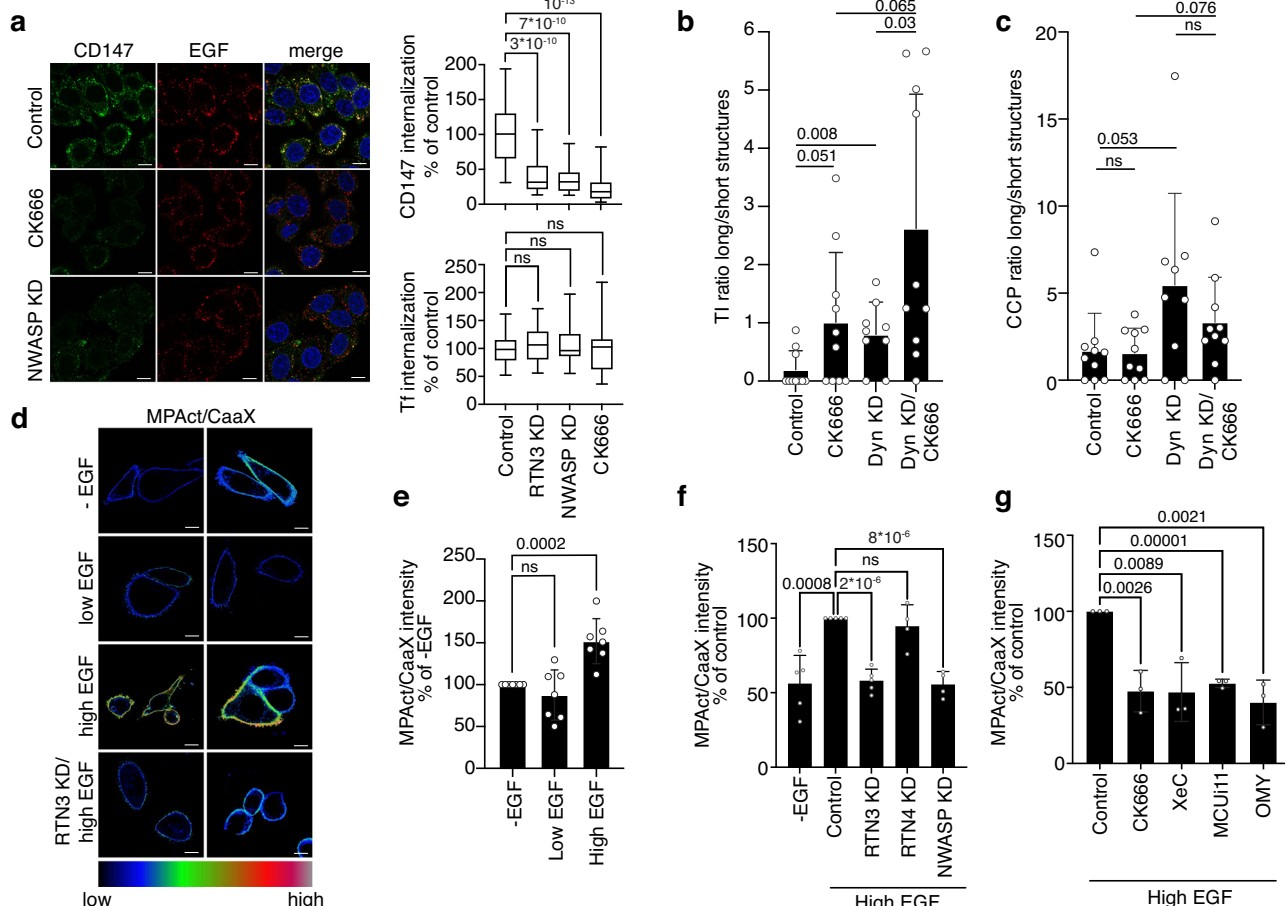

**Fig. 6 | Ca²⁺ and ATP are required for cortical actin cytoskeleton remodeling needed for the fission of NCE Tis. a** CD147 internalization was monitored in HeLa cells subjected to N-WASP-KD, RTN3-KD or treatment with CK666 and stimulated with high dose EGF. Left, representative IF images, CD147 (green), EGF (red), DAPI (blue), Bar, 10 μm. Right upper, quantification of relative CD147 fluorescence intensity expressed as % of control. *N* = number of cells: Control/N = 338 (*n* = 4); RTN3 = KD/N = 158, NWASP-KD/N = 237, CK666/N = 250 (*n* = 3). Right lower, Alexa647-Tf internalization was monitored for 8 min at 37 °C in HeLa cells treated as above. The data reported for RTN3 KD are the same as displayed in Fig. 4b. N=number of cells: Control/N = 246 (*n* = 6); RTN3-KD/N = 185, NWASP-KD/N = 276, CK666/N = 295 (*n* = 3). Box plots are defined as in Fig. 2f. **b, c** EM morphometric analysis of the length of EGFR gold-positive TIs (**b**) and CCPs (**c**) in HeLa cells subjected to dynamin KD (Dyn-KD) and/or treated with CK666. Data are expressed as Fig. 4d, mean ± SD; N = cell profiles. Control/N = 10, CK666/N = 10, Dyn-KD/N = 9, Dyn-KD/CK666/N = 10 (**b**); Control/N = 10, CK666/N = 10, Dyn-KD/N = 9, Dyn-KD/CK666/N = 10 (c); a representative experiment of *n* = 2 is shown. **d** Ratiometric

analysis of membrane proximal actin (MPA) density. HeLa cells were subjected or not to RTN3 KD, followed by MPAct-mCherry and YFP-CaaX co-transfection, and stimulated with high or low EGF or left unstimulated (-EGF). Left, two representative images of the ratiometric analysis of MPA density (MPAct signal normalized over the CaaX PM marker). Bar, 10 μm. **e** Right, mean raw integrated fluorescence intensity ±SD is reported as a percentage relative to control cells. *N* = number of cells: -EGF/N = 424, LowEGF/N = 495, HighEGF/N = 568 (*n* = 7). **f, g** Ratiometric analysis of MPA density as in **d**, in HeLa cells subjected to the indicated KDs (**f**) or inhibitor treatments (**g**) and stimulated with high dose EGF. Data are expressed as mean ± SD. Inhibitors were present during the stimulation. *N* = number of cells: (**f**) -EGF/N = 316/n = 5, Control/N = 356/n = 5, RTN3-KD/N = 409/n = 5, RTN4-KD/N = 311/n = 4, NWASP-KD/N = 370/n = 4; (**g**) Control/N = 306, CK666/N = 228, XeC/N = 231, MCUi11/N = 232, OMY/N = 189 (*n* = 3). Exact *p*-values (Each pair Student's *t* test, two-tailed) are shown in all panels; ns, not significant; n=biological replicates. Source data are provided as a Source Data file.

and/or inhibition of RTN3, IP3R, MCU, or mitochondrial ATP synthase, while RTN4 KD and AP2 KD had no effect (Fig. 6f, g and Supplementary Fig. 6f). These data support a model where NCE requires the interplay between the PM, ER and mitochondria to generate Ca²⁺ waves and localized mitochondrial ATP production, which are in turn needed for cortical actin remodeling that cooperates with dynamin in the fission of NCE TIs.

We also investigated the role of the motor protein dynein, which was previously involved in the fission of TIs in the NCE of Shiga toxin[37]. The inhibition of dynein with Ciliobrevin D decreased CD147/EGF NCE, but not Tf CME (Supplementary Fig. 7a, b). Dynein probably acted at the initial steps of NCE, as its inhibition reduced ER-PM contact sites and TI formation. (Supplementary Fig. 7c, d). The involvement of dynein in the early steps of NCE might be linked to its role in cortical ER motility[38]. Our data do not exclude, however, an involvement of dynein

also at the fission step in concert with actin and dynamin, as previously shown for the NCE of Shiga toxin.

## EGFR signaling and mitochondrial activity at tripartite contact sites are required for EGF-induced cell migration

We asked whether the EGF-induced multiorganelle platform has implications for EGFR signaling and biological output, focusing on collective migration which depends on functional NCE[39]. Given the relevance of EGFR-NCE in keratinocytes, we used human HaCaT cells, since they are extensively used as a model for cell migration and wound healing induced by growth factors[40].

First, we confirmed that the EGFR-NCE mechanism is conserved in HaCaT cells, as demonstrated by the RTN3/IP3R/MCU-dependent internalization of CD147 and EGF after stimulation with high dose EGF (Fig. 7a and Supplementary Fig. 8a–c). Tf internalization was also used

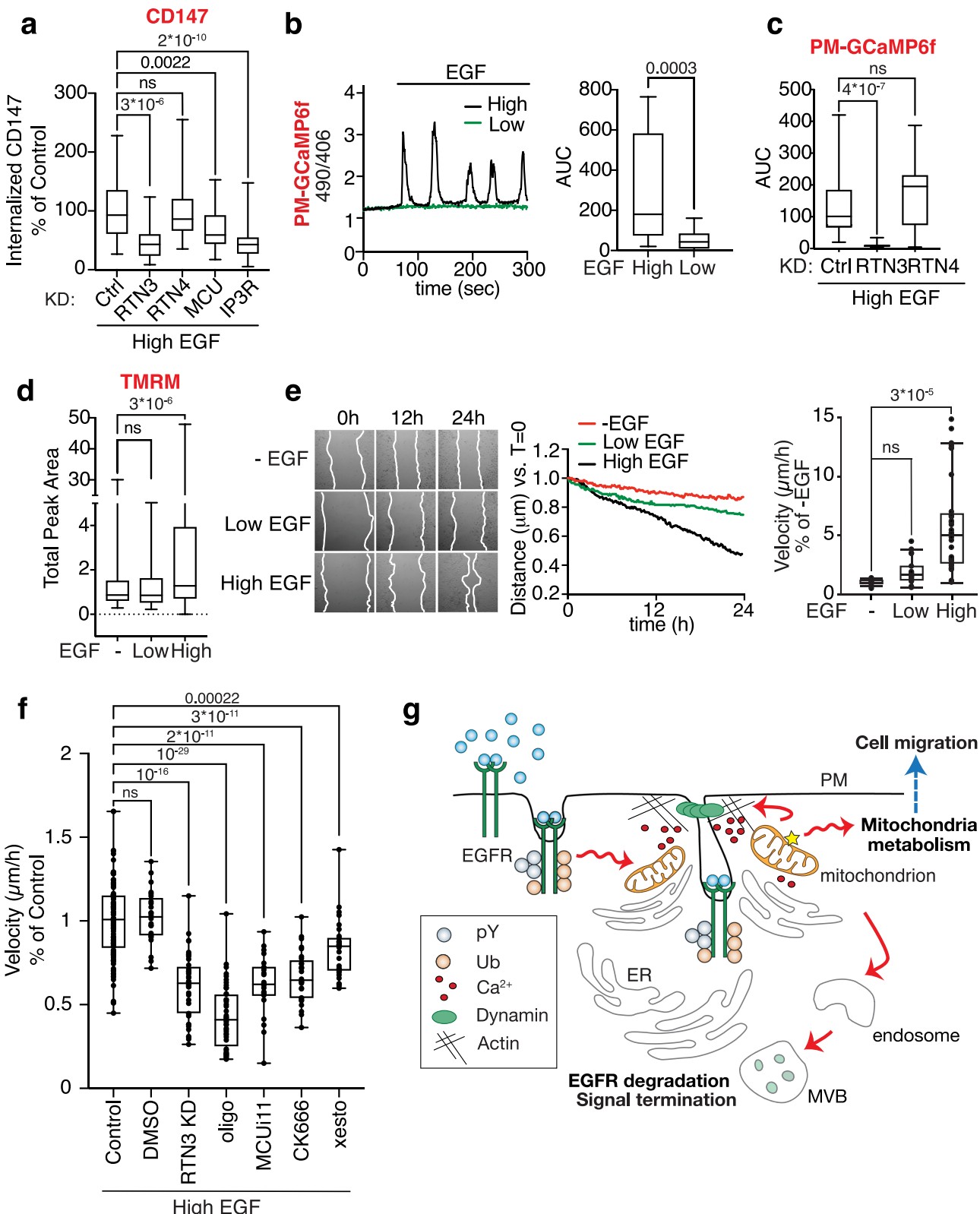

as a negative control (Supplementary Fig. 8d). We also verified the generation of RTN3-dependent PM localized $Ca^{2+}$ waves (Fig. 7b, c and Supplementary Fig. 8e) and an increase in $\Delta\Psi m$ at high dose EGF in these cells (Fig. 7d and Supplementary Fig. 8f).

In a wound healing assay, high EGF treatment significantly increased cell migration velocity compared to low EGF which behaved like the serum starved control (Fig. 7e and Supplementary Movie 6). This increased migration velocity was dependent on RTN3, ER-mediated $Ca^{2+}$ release, mitochondrial buffering activity, mitochondrial ATP production and actin polymerization, as demonstrated by RTN3 KD or inhibitor treatments (Fig. 7f). Thus, the NCE tripartite organelle platform controls previously unknown aspects of the EGFR

**Fig. 7 | The EGFR-NCE multiorganelle platform mediates EGF-induced migration in keratinocytes. a** CD147 internalization in HaCaT cells subjected to the indicated KDs or mock transfection (Ctrl). Internalized CD147 was measured by IF after stimulation with high dose Alexa647-EGF for 12 min at 37 °C, as described in Fig. 4a. Mean integrated fluorescence intensity is reported as a percentage of the control cells. N = number of cells: Control/N = 301/n = 6, RTN3-KD/N = 97/n = 3, RTN4-KD/N = 179/n = 5, MCU-KD/N = 221/n = 4, IP3R-KD/N = 210/n = 4. **b** HaCaT cells expressing the $Ca^{2+}$ sensor PM-GCaMP6f were stimulated with high or low EGF. Left, Representative single-cell $Ca^{2+}$ response curves presented as the ratio of fluorescence emission at 490/406 nm. Right, AUC of the $Ca^{2+}$ response. N=number of cells: High EGF/N = 17, Low EGF/N = 20 (n = 2). **c** $Ca^{2+}$ response in HaCaT cells subjected to RTN3 or RTN4 KD or to mock transfection (Ctrl) and stimulated with high EGF dose as in **b**. AUC is shown. N = number of cells: Control/N = 18, RTN3-KD/N = 25, RTN4-KD/N = 14. A representative experiment of three independent replicates is shown. **d** ΔΨm in HaCaT cells stimulated or not with low or high EGF, measured using TMRM. Total peak area is reported. N=number of cells:

-EGF/N = 206, Low EGF/N = 253, High EGF/N = 327. A representative experiment of n = 2 is shown. **e** Wound healing in sub-confluent HaCaT cells stimulated or not with low or high EGF was monitored by time-lapse video microscopy. Left, representative images at the indicated time points. Middle, distance covered by cells in the different conditions corrected to the baseline at t = 0. Right, wound closure velocity (µm/h) at t = 24 h is shown. N=number of cells: -EGF/N = 15, Low EGF/N = 23, High-EGF/N = 34 (n = 3). **f** Wound closure velocity (µm/h) in sub-confluent HaCaT cells stimulated with high EGF dose and subjected to RTN3 KD or the indicated treatments for 24 h. N = number of cells: Control/N = 98/n = 6, DMSO/N = 24/n = ), RTN3-KD/N = 40/n = 2, OMY/N = 44/n = 3, MCUi11/N = 22/n = 2, CK666/N = 28/n = 2, XeC/N = 30/n = 2). **g** Schematic showing the crosstalk between EGFR-NCE, ER, and mitochondria in controlling EGFR endocytosis, receptor fate and cell migration. Box plots in all panels are defined as in Fig. 2f. Exact p-values (each pair Student's t-test, two-tailed) are shown; ns not significant, n biological replicates. Source data are provided as a Source Data file.

signaling response that impinge on mitochondrial ATP metabolism and collective cell migration.

It remains to be established how the NCE platform, which is formed at early time points after EGF stimulation, could support a long-term response such as cell migration. One possible explanation is that a small population of active EGFRs, continuously recruited to the NCE platform, is responsible for sustaining migration. In agreement with this possibility, we found that a residual fraction of phosphorylated receptors and downstream signaling effectors persist upon long-term EGF exposure (Supplementary Fig. 9a). Moreover, we show that the migratory phenotype still depends on EGFR kinase activity even at late time points (Supplementary Fig. 9b), arguing that continuous EGFR-dependent signaling is needed, possibly through the NCE platform. It remains to be established whether the signal is emanated at the PM or en route in the endosomal compartment.

## Discussion

Interorganelle contact sites are critical to many cellular physiological processes since, by controlling the transfer of ions, proteins and lipids between organelles, they regulate organelle dynamics and intracellular signaling[41–44]. Herein, we show that tripartite contact sites involving the PM, the ER and mitochondria, are transiently formed upon stimulation of cells with high doses of EGF. These structures are involved in at least two EGF-dependent functions (Fig. 7g): i) they execute the-process of EGFR-NCE which leads to receptor degradation and signaling attenuation[5]; ii) they act as signaling platforms to induce collective cell migration. This latter finding is relevant to the pleiotropic cellular phenotypes induced by EGF, which acts as a potent mitogen when used at low concentrations[45–47], whereas at higher doses, it elicits a collective migratory response [REF[39,48] and this paper]. Our findings shed light on how a quantitative signal (i.e., ligand concentration) is deconvoluted into specific cellular responses through the dose-dependent formation of a multiorganelle signaling platform.

A complex cascade of events takes place at the multiorganelle platform. Initially, EGF (at high doses only) triggers the formation of ER-mitochondrial contacts in proximity of the PM. High doses of EGF also induce the release of $Ca^{2+}$ from the ER through IP3R, close to the PM[5]. Since it has been demonstrated that these $Ca^{2+}$ waves can stimulate the association of the ER with mitochondria[49,50], it is very likely that the formation of the tripartite complex is further stabilized by EGF through this mechanism.

Mechanistically, RTN3 is required for the formation of this organelle platform, as its ablation affects the proximity of ER-mitochondria to the PM. These findings can be interpreted according to two different scenarios. RTN3 may participate exclusively to the formation of PM-ER contacts[5], with consequent distancing of the ER-Mitochondria complex from the PM, upon RTN3-KD. Alternatively, RTN3 might be

additionally involved in the formation of ER-Mitochondria contacts. Further work will be required to resolve this issue.

The ablation of RTN4 did not phenocopy the RTN3 KD. This is interesting, since RTN3 and RTN4 belong to the same family of ER-resident factors and share a localization in the cortical tubular ER, through the reticulon homology domain (RHD). However, while RTN4 is directly involved in shaping ER tubules[5,9,51], emerging evidence suggests that RTN3 plays a more specific role in recognizing highly curved ER membrane domains and in facilitating the assembly of factors involved in regulating membrane contact sites at these locations without directly influencing ER morphology[5,9,52]. This might be due to the unique cytosolic domains of the two proteins, which likely contribute to their diverse functions associated with tubular ER. Whatever the case, our data argue that the effects of RTN3 KD are likely attributable to the disruption of a specific subset of membrane contact sites rather than to a general impairment of ER morphology.

ER contacts with EGFR-positive endosome/MVBs have been also documented[9,53–55]. Whether the formation of these contacts involves mechanisms similar to those underlying the ER-PM contacts herein described remains to be established. Notably, the long isoform of RTN3 has been implicated in the tethering of ER tubules with EGFR-positive endosomes[9], we have previously shown that the short isoform of RTN3 is sufficient to rescue the NCE internalization defect upon RTN3 KD[5]. This raises the possibility that specific RTN3 isoforms are involved at different steps of the endocytic route possibly interacting with specific sets of tethering factors.

We acknowledge that defining the precise molecular mechanisms of membrane tethering is a relevant question; it should be appreciated, however, that this will require a substantial experimental effort, including: (i) defining whether the interaction between EGFR and RTN3 is direct or indirect, (ii) if indirect, identifying the intervening proteins, (iii) if direct, mapping the surface of EGFR/RTN3 interaction, (iv) generating mutants and performing reconstitution assays in the appropriate KD background, and (v) elucidating the involvement of other accessory tethering factors in contact formation. Further experiments are required to elucidate these points.

At the functional level, once the organelle platform has been assembled, we show that the recruited mitochondria perform a dual role. On the one hand, they permit the creation of $Ca^{2+}$ oscillations, in keeping with their known $Ca^{2+}$ buffering activity[15–17]. This ensures that intermittently high $Ca^{2+}$ concentrations are maintained in the vicinity of the platform, preventing the feedback closure of the IP3R channel by $Ca^{2+}$[56]. On the other hand, the consequent $Ca^{2+}$ oscillations inside the mitochondria are sensed by the mitochondrial Krebs cycle enzymes and ATP synthase machinery, increasing NAD(P)H/FAD and ATP production[21]. ATP released in the microenvironment of the platform is initially required for TI formation, while $Ca^{2+}$ and ATP are both needed at a later stage to complete EGFR-NCE, via the induction of

cortical actin cytoskeleton remodeling, which, together with dynamin, is required for the fission step of NCE-TIs (see proposed model in Fig. 7g). The same chain of events appears to be required for collective cell migration, which is affected by the inhibition of mitochondrial $Ca^{2+}$ buffering activity and mitochondrial ATP synthase. We note, however, that the chronic inhibition of mitochondrial buffering activity and energetics. employed in our experiments, might have broad effects on cell behavior that go beyond NCE and the EGF-dependent local release of ATP.

There is vast availability of ATP in the cell (in mM concentrations), so why is a burst of ATP at the PM needed for PM invagination, cortical actin remodeling, and vesicle fission? One explanation could be the higher energy requirement and the increased consumption of ATP required for the execution of the EGFR-NCE program compared to other endocytic programs, e.g., EGFR-CME. Indeed, in the first phase of EGFR-NCE, the formation of ER-PM-mitochondria contacts is needed to allow PM invagination and TI elongation, two energy-demanding processes[57,58]. In agreement, we observed: i) a requirement of mitochondrial ATP for TI formation (Supplementary Fig. 5d) and ii) a decrease in cytosolic ATP upon high dose EGF stimulation (Supplementary Fig. 3c, d). In a second phase of EGFR-NCE, an extra burst of ATP provided by the proximal mitochondria is necessary to induce Arp2/3 relocalization at the PM and cortical actin polymerization, required to complete the fission event, in concert with dynamin. In contrast, CME differs in its energy demands. In the first step of membrane bending and vesicle formation, the clathrin triskelion self-assembles into a polyhedral coat surrounding the membrane in a self-sustaining process that does not require ATP[57,59]. This leaves the cytosolic pool of ATP available for the fission of CCPs operated by dynamin.

Similar considerations may apply to the migratory phenotype. Mitochondria are involved in the control of cell migration through their role in $Ca^{2+}$ signaling and energy production[60–62]. Moreover, there is evidence that mitochondrial movement can match energy demands throughout the cell[63,64] and that repositioning of mitochondria at cell periphery is needed for cancer cell migration and invasion[65–68].

One final question is why an effector (migration) and an attenuation (internalization/degradation) mechanism are integrated at the tripartite structures. When we originally discovered that EGFR-NCE is primarily coupled to receptor degradation, as opposed to CME that is coupled to receptor recycling and signaling sustainment, we proposed that its major function could be to protect the cell from overstimulation[5,8]. Our present findings highlight a more complex picture where the same signaling platform appears to enact a specific concentration-dependent effector phenotype while ensuring its termination when no longer needed.

Importantly, we show here that another growth factor receptor, HGFR/MET, exploits the same NCE mechanism and activates the same circuitry influencing calcium and mitochondrial energetics as the EGFR. Given the vast roles that EGFR and HGFR signaling plays in the physiology of epithelial cells (e.g., ranging from the regulation of proliferation and migration to controlling stem cell homeostasis) and the frequent aberrant regulation of these signaling pathways in human cancers, the discovery of a mechanism that links their signaling to mitochondrial energetics could provide insights into novel therapeutic approaches combining EGFR/HGFR- and mitochondria-targeted therapies.

## Methods

### EGF concentrations and reagents
Throughout the manuscript low and high dose EGF means 1 and 100 ng/ml, respectively. Alexa-labeled EGF was used at a concentration of 1 μg/ml of the conjugated species, corresponding to an actual EGF concentration of ~40 ng/ml. EGF was from PeproTech; Alexa-EGF and Alexa-Tf were from Molecular Probes. Purified human recombinant HGF (100-250 ng/ml, as indicated) was obtained from R&D Systems. Protein-A Gold 10 nm was from Utrecht University; EM grade glutaraldehyde and paraformaldehyde were from Electron Microscopy Sciences; ruthenium red and secondary rabbit anti-mouse were from Sigma. Fluorescent secondary antibodies were from Jackson ImmunoResearch. Beetle Luciferin Potassium Salt for Luciferase-based assays was from Promega. Oligomycin (OMY), histamine, and succinate were from Sigma, xestospongin C (XeC) was from abcam, CK666 and bongkrekic acid were from Merck Life Technology S.R.L., MCUi11 was from CliniSciences and Gefitinib was from Merck Life Technology S.R.L., Ciliobrevin D was from TargetMol (T14965). Information about antibody brands and conditions of usage can be found in Supplementary Table 1.

### Drug treatments
Cells were treated with 20 μM XeC for 30 min (except for the wound-healing assay where it was used at 2 μM for 24 h), 50 μM CK666 for 1 h, 1 μM OMY for 15 min (except for the PM-Luc assay where it was used at 10 μM for 40 min) and 15 hours for the evaluation of intracellular ATP content, 50 μM MCUi11 for 30 min, 50 μM bongkrekic Acid for 15 min, 5 μM Gefitinib for different time points (6, 12, 18, 48 hours), 50 μM Ciliobrevin D for 20 mins of pre-treatment (and then kept for all the duration of the experiments, CD147/EGF/TF internalization and EM). The drugs were kept for the duration of the different experimental procedures (CD147 internalization followed by EM/IF analysis, TMRM live recording, PM-Luc ATP recording, MPAct/CaaX actin polymerization, and wound healing).

### Constructs
pGP-CMV-GCaMP6f was a gift from Douglas Kim & GENIE Project (Addgene plasmid # 40755; http://n2t.net/addgene:40755; RRID:Addgene 40755). For the SNAP25-GCaMP6f tagged version, the SNAP25-Aequorin sequence was used as a template for the PCR reaction. To generate PM-Luc stably expressing clones, SNAP25 tag was subcloned from SNAP25-GCaMP6f in the pcDNA3.1 vector (Invitrogen) by PCR using XHOI and BAMHI restriction enzymes. Luciferase cDNA was subcloned by PCR from the CytLuc-pGL3 basic plasmid into the SNAP25-pcDNA3 vector via Nebuilder Hifi DNA Assembly Cloning Kit using the XHOI restriction enzyme. Then, SNAP25-luciferase cDNA was subcloned into the pLVX-Puro vector (Clonetech) using the NEBuilder HiFi DNA Assembly Cloning Kit and the BAMHI restriction enzyme. To generate the PM-GCaMP6f stably expressing HeLa and HaCaT clones, the SNAP25-GCaMP6f cDNA was subcloned from SNAP25-GCaMP6f in the pEN_TmiR vector and recombined in the pSLIK-neo lentiviral vector[69] using the Gateway System (Invitrogen). HeLa and HaCaT cells were then infected with the pSLIKneo-SNAP25-GCaMP6f vector and selected in a medium containing 400 μg/ml neomycin for seven days. CytoLuc-pGL3 basic (Promega #E1751), pLVX-NEO-CMV-2mt-GCaMP6s, and SNAP25-Aequorin were kindly provided by Dr. Massimo Bonora and Dr. Paolo Pinton from the University of Ferrara[13,70]. C1-MPAct-mCherry, CFP-CaaX, and YFP-CaaX were gifts from Tobias Meyer (Addgene plasmids #155222, #155232, #155233; http://n2t.net/addgene:155222; RRID:Addgene_155222).

### Cell culture and transfection
HeLa cells (human cervix epithelial cell line, an isolate from IEO Institute)[7] were cultured in Minimum Essential Medium (MEM, Sigma) supplemented with 10% FBS, 2 mM glutamine, sodium pyruvate 1 mM (Euroclone), 0.1 mM non-essential amino acids (Euroclone). The pLVX-SNAP25-Luciferase HeLa clone was grown in the same medium as HeLa cells supplemented with 1 μg/ml Puromycin (Adipogen Life Sciences). The pSLIK-SNAP25-GCaMP6f and pLVX-mitoGCaMP6s HeLa clones were grown in the same medium as HeLa cells supplemented with 750 μg/ml G418 (Adipogen Life Sciences). HaCaT (human immortalized keratinocytes, CLS Cell Lines Service, 300493) and HeLa OSLO (a HeLa

isolate kindly provided by Prof. IH Madshus, University of Oslo, Norway[7] cells were cultured in DMEM medium (ThermoFisher) supplemented with 10% FBS and 2 mM glutamine. All cells were cultured at 37 °C and 5% CO$_2$. Cells were passaged every 2-3 days to maintain subconfluency. HeLa cells at 50 – 60% confluence were transfected using FuGene (Promega) according to Manufacturer's instruction. Experiments were performed 24–48 h after transfection.

All human cell lines were authenticated at each batch freezing by STR profiling (StemElite ID System, Promega). All cell lines were tested for mycoplasma at each batch freezing by PCR (32) and biochemical assay (MycoAlert, Lonza).

### RNA interference
RNAi was performed with Lipofectamine RNAimax (Invitrogen), according to the manufacturer's protocol: for RTN3 KD, RTN4 KD, NWASP KD, and clathrin heavy chain KD, cells were transfected with 2 cycles (on day 1 in suspension, and on day 2 in adhesion) of 8 nM of oligos; for AP2μ1 KD, MCU KD and IP3-R1, IP3-R2, IP3-R3 KDs, 1 cycle of 8 nM of oligos; for Dynamin 1 and 2 KD, 1 cycle of 4 nM of oligo. Mock-treated cells (incubated with Lipofectamine RNAimax in optimum medium) were used as control. Experiments were performed 72 h after the second transfection. The Oligo brand and sequence are listed in Supplementary Table 2.

### Western blotting and antibodies
Cells were lysed by adding RIPA buffer (50 mM Tris–HCl, 150 mM NaCl, 1 mM EDTA, 1% Triton X-100, 1% sodium deoxycholate, 0.1% SDS), plus protease inhibitor cocktail (CALBIOCHEM) and phosphatase inhibitors (20 mM sodium pyrophosphate pH 7.5, 50 mM NaF, 2 mM PMSF, 10 mM Na$_3$VO$_4$ pH 7.5) directly to cell plates and lysate was clarified by centrifugation at 17,000 × $g$ for 20 min at 4 °C. Protein concentration was measured by Bradford Assay (Biorad) and 20–50 μg of protein was run on 4-20% gradient pre-cast Gels (Biorad) and transferred using Trans-Blot (Biorad) according to manufacturer's instructions. Filters were blocked with 5% milk diluted in TBS 0.1% Tween20 (TBS-T) and then incubated overnight with primary antibody according to the datasheet. Following 3 washes with TBS-T, filters were then incubated with the appropriate secondary antibody conjugated with horseradish peroxidase. After 3 more washes, the signal was detected at Chemidoc (Biorad) using ECL from Amersham or Biorad.

### CD147/EGF/Tf – HGFR/HGF internalization assays
For antibody internalization assays, serum-starved cells were first incubated with anti-CD147 antibody for 60 min at 4 °C and then with an Alexa-488 fluorescent secondary antibody (in green) for 30 min at 4 °C. Cells were then stimulated with high dose Alexa-conjugated EGF (see section "EGF concentration") at 37 °C for 5 min in HeLa cells and 12-15 min for HaCaT cells, or with high dose unlabeled HGF (1 μg/ml, preactivated for 20 min at room temperature followed by 15 mins at 37°C in 100% serum, then diluted in serum-starved medium for obtaining a final concentration of 100 ng/ml in 10% serum prior to use) for 8 min. As a control, we also incubated CD147 in the absence of the ligand (in the case of HGF experiment, in the presence of 10% serum) and we observed no detectable CD147 internalization. After internalization, cells were acid wash-treated (100 mM Glycine-HCl) pH 2.2, fixed in 4% paraformaldehyde and processed for IF. Images were obtained using a Leica TCS SP8 confocal microscope equipped with a 63X oil objective and processed using ImageJ. An ad hoc designed macro was used to quantify CD147 signal upon stimulation with EGF[5]. CD147 signal was highlighted applying an intensity-based threshold (Default method), and then fluorescence intensity per field was calculated using the "Measure" command, limiting measurements to threshold. This value was then divided by the number of nuclei in the field, counted using the DAPI signal, to calculate the CD147 fluorescence intensity per cell.

For HGFR internalization assay, HeLa cells were stimulated with high dose HGF at 37 °C for 8 min as described above. Following fixation, HGFR protein was stained with anti-HGFR antibody recognizing the extracellular domain followed by CY3-labeled secondary antibody before permeabilization (referred to as PM HGFR). After staining, the cells were further fixed and then permeabilized to allow for another round of staining with anti-HGFR antibody followed by Alexa 647-labeled secondary antibody (referred to as Total HGFR). The resulting images were analyzed using ImageJ software. The PM HGFR signal was selectively emphasized by applying an intensity-based threshold using the "Otsi" method. The fluorescence intensity per field was then calculated using the "Measure" command, with measurements limited to the thresholded region. A similar analysis was performed for the Total HGFR signal using the "Moments" method. Finally, the ratio of PM HGFR to Total HGFR was used to normalize the levels of PM HGFR in each experimental condition.

To quantify internalized EGF/Tf, cells were stimulated with high dose Alexa-conjugated EGF or Alexa-conjugated Tf (see section "EGF concentration") at 37 °C for 5 min (EGF) or 8 min (TF) in HeLa cells, and 12 min for both ligands in HaCaT cells. Before fixation, samples were subjected to acid wash treatment to visualize only internalized ligand. Images were analyzed using ImageJ. EGF signal was highlighted applying a 10-intensity based threshold (Default method), and then fluorescence intensity per field was calculated using the "Measure" command, limiting measurement to threshold. This value was then divided by the number of nuclei in the field, and counted using the DAPI signal to calculate the EGF fluorescence intensity per cell.

### Immunofluorescence for evaluating mitochondrial morphology
HeLa cells underwent either RTN3 knockdown (KD) or mock transfection before being plated on 13-mm coverslips. Subsequently, the cells were treated with 1 nM Mitotracker CMXROS (Life Technologies Italia, M7512) for 30 min at 37 °C in a complete medium, followed by two washes with PBS. Then, cells were either stimulated with high EGF (100 ng/ml) or left unstimulated, followed by fixation with 4% paraformaldehyde for 8 min at room temperature, staining with DAPI for nuclei detection, and mounting on glass slides using glycerol mounting media. Images were acquired using a Spinning Disk Confocal microscope equipped with a 100X objective, with serial z-stack acquisitions taken every 300 nm to better assess mitochondrial network complexity. Mitochondrial morphology, including mean volume, circularity, and solidity, was analyzed using ImageJ with a custom macro. Results are expressed as percentage relative to unstimulated control cells. Statistical analysis was conducted using Graph Prism software.

### Electron microscopy
**Pre-embedding immunolabeling.** Serum-starved cells were incubated with anti-EGFR 13A9 (Genetech) or with anti-CD147 antibody, followed by incubation with rabbit anti-mouse, and, finally, with Protein-A Gold 10 nm (30 min incubation on ice/each step). Cells were then incubated at 37 °C for 5 min with 30 ng/ml EGF, as indicated. A control sample left at 37 °C for 5 without EGF was included in the experiment to control that no internalization was induced by the antibody in the absence of ligand. Cells were then washed in PBS and fixed for 1 h at room temperature in 1.2% glutaraldehyde in 66 mM sodium cacodylate buffer pH 7.2 containing 0.5 mg/ml of ruthenium red. After quick washes with 150 mM sodium cacodylate buffer, the samples were post-fixed in 1.3% osmium tetroxide in a 66 mM sodium cacodylate buffer (pH 7.2) containing 0.5 mg/ml ruthenium red for 2 h at room temperature. Cells were then rinsed with 150 mM sodium cacodylate, washed with distilled water and enbloc stained with 0.5% uranyl acetate in dH$_2$O overnight at 4°C in the dark. Finally, samples were rinsed in dH$_2$O, dehydrated with increasing concentrations of ethanol, embedded in Epon and cured in an oven at 60°C for 48 h.

Ultrathin sections (70–90 nm) were obtained using an ultramicrotome (UC7, Leica microsystem, Vienna, Austria), collected, stained with uranyl acetate and Sato's lead solutions, and observed in a Transmission Electron Microscope Talos L120C (FEI, Thermo Fisher Scientific) operating at 120 kV. Images were acquired with a Ceta CCD camera (FEI, Thermo Fisher Scientific).

**Morphometry of NCE TIs and of CCPs.** Morphometry was performed as previously described[5]. Briefly, cellular profiles of thin sections of cells immunolabeled with anti-EGFR antibody (13A9) and stained with ruthenium red were acquired. For the quantification of the number of CCPs or TIs upon different treatment, gold particle clusters present in PM connected (ruthenium red-positive) structures of randomly selected cells were acquired at a nominal microscope magnification of ×22,000 (pixel size 0.6 nm). Gold clusters were assigned to one of the two categories based on the presence of a clathrin coat and the distance between PM and the tip of the invagination was measured using ImageJ. The number of ruthenium red-positive endocytic structures identified was divided by the PM length measured with ImageJ on acquired low magnification micrographs (nominal microscope magnification of ×1200) and expressed as number of structures per 100 mm of PM.

For quantification of the elongation of endocytic structures the number of short structures were divided by the number of long ones (respectively > 200 nm for CCPs and >of 300 nm for Tis). Tis structures shorter than 150 nm or structures that cannot be unequivocally ascribed to one or the other category were excluded from the analysis.

**KDEL-HRP/DAB visualization of ER.** Cells were fixed in 1% glutaraldehyde in 0.1 M sodium cacodylate buffer (pH 7.4) for 30 min at room temperature, followed by incubation with 0.3 mg/ml 3,3'-diaminobenzidine tetrahydrochloride (DAB) and 0.03% hydrogen peroxide in 0.1 M sodium cacodylate buffer pH 7.4 for 20 min at room temperature. Samples were rinsed in sodium cacodylate buffer and post-fixed with 1.5% potassium ferrocyanide, 1% osmium tetroxide in 0.1 M sodium cacodylate buffer (pH 7.4) for 1 h on ice. After enbloc staining with 0.5% uranyl acetate in $dH_2O$ overnight at 4°C in the dark, samples were dehydrated with increasing concentrations of ethanol, embedded in Epon and cured in an oven at 60°C for 48 h. Ultrathin sections (70–90 nm) were obtained using an ultramicrotome (UC7, Leica microsystem, Vienna, Austria), collected, stained with uranyl acetate and Sato's lead solutions, and observed in a Transmission Electron Microscope Talos L120C (FEI, Thermo Fisher Scientific) operating at 120 kV. Images were acquired with a Ceta CCD camera ((FEI, Thermo Fisher Scientific).

**Serial section electron tomography.** Serial thick sections (130–150 nm) were collected on formvar-coated copper slot grids. Gold fiducials (10 nm) were applied on both surfaces and the grids were stained with 2% methanolic uranyl acetate and Sato's lead citrate. The samples were imaged using a 120 kV Talos L120C (Thermo Fisher Scientific). Tilted images (+65/−65 according to a Saxton scheme) were acquired with Tomography 4.0 acquisition software (Thermo Fisher Scientific) using a 4kx4k Ceta16M camera (Thermo Fisher Scientific). Tilted series alignment, tomography reconstruction, and serial tomograms joining were performed using the IMOD software package (Mastronarde, 1997).

**3D ER-mitochondria visualization and quantification.** Tomograms were semi-automatic segmented using Microscope Image Browser (MIB) (DOI: 10.1371/journal.pbio.1002340). ER stained by KDEL-HRP/DAB electron dense precipitate was segmented using MIB Frangi Filter; mitochondria and PM were manually segmented with the brush tool on few slices and automatically interpolated through the 3D stacks. For visualization, the MIB model was exported to IMOD. For quantification, the segment profiles were exported to MatLab (MathWorks Inc.) and the distances between ER and mitochondria profiles were calculated with a MatLab script. Areas of contact ( < 20 nm) with mitochondria were highlighted in red and contacts with a red area/mitochondrion (>0.05) was quantified.

**ER-Mitochondria contact site analysis by CLEM.** HRP-KDEL[5,71] transfected cells were fixed with 4% paraformaldehyde, 0.01% Glutaraldehyde in 0.1 M HEPES pH 7.4. The ER was then visualized incubating the cells using fluorescent tyramide (TSA-FITC 1:200, AKOYA Biosciences), 0.003 $H_2O_2$ and 0.05% Tween-20 in 100 mM borate buffer. Cells were then permeabilized with 0.1% saponin and stained with TOMM20 antibody (NBP1-81556 from Novus Biologicals) visualized with a secondary antibody conjugated with Alexa564 (Invitrogen). Samples were then dehydrated and embedded in LR white resin using microwave irradiation. Ultrathin sections (100 nm) were obtained with a Leica UC7 ultramicrotome, collected on glass coverslips and nuclei were stained with Hoechst. The fluorescence signal was collected in a DeltaVison widefield microscope (DeltaVision Elite; Applied Precision/ GE Healthcare) using a 100x plan-apo oil immersion objective. The acquired 3D stack was deconvoluted using software (DeltaVision Elite; Applied Precision/GE Healthcare) and analyzed with ImageJ software. ER-mitochondria "contacts" were identified with the object-based colocalization function of JaCop Plugin[72]. To quantify juxtamembrane contacts using the Hoechst channel, nuclei and the cell borders were manually segmented and a relative distance map was created having a value of 0 close to the nucleus and 1 close to the cell border. Contacts in proximity to the PM (relative distance map value higher than 0.75) were identified using ImageJ Analyze particles plugin.

To validate the quantification procedure used the CLEM approach evaluating whether the colocalization regions identified with ImageJ corresponded to a real contact site at EM level. To make the fluorescent section compatible with EM imaging we introduced in the embedding procedure heavy metal staining. Briefly, after KDEL-HRP/ TSA staining and TOMM20 immunolabeling, cells were post-fixed with 0.2% osmium tetroxide, 0.3% potassium ferrocyanide in 0.1 M cacodylate buffer for 5 min with microwave irradiation (two microwave pulses of 40 s over 5 min). Then samples were further contrasted with 0.5% uranyl acetate for 1 min in the microwave and finally dehydrated and embedded in LR white resin. This procedure confers to the sample enough contrast enabling the visualization of the cellular membrane during EM imaging but decreased the fluorescent signal. Therefore, for the CLEM approach, we used mitotracker staining for mitochondria visualization since it gave a brighter signal compared to TOMM20 immunostaining. Ultrathin sections (100 nm) were obtained with a Leica UC7 ultramicrotome, collected on glass coverslips, and nuclei were stained with Hoechst and imaged with a DeltaVision widefield microscope. After being imaged, the sections were detached from the glass coverslips using 1% hydrofluoric acid and collected on formvar-carbon coated slot grids. The same cell acquired in the fluorescence microscope was then identified at the EM level and images were collected with a Talos L120C transmission electron microscope. Fluorescent and EM images were aligned with ec-clem ICY plugin, and the number of contact sites revealed by EM (20 nm distance) and fluorescence (JaCoP plugin) were evaluated.

**ER-PM contact site analysis by EM.** ER-PM contact analysis was performed as previously described[5]. Briefly, For ER labeling, cells were transfected with HRP-KDEL; 24 hours after transfection, cells were subjected to pre-embedding immunolabeling with anti-CD147 antibodies as described above. Cells were then fixed and processed for EM analysis. To quantify the proximity of ER and CD147, random images of CD147 positive structure were acquired at a nominal magnification of 22 K. CD147 positive structures were considered associated or in proximity when at least an ER tubule was present at ≤ 20 nm.

## Measurements of intracellular Ca²⁺ concentration

**Aequorin measurements.** HeLa cells grown on 13-mm-round glass coverslips at 50% confluence were transfected with the appropriate PM-targeted aequorin chimeras. Aequorin constructs and protocols were previously described[20]. All aequorin measurements were performed in KRB buffer (135 mM NaCl, 5 mM KCl, 0.4 mM $KH_2PO_4$, 1 mM $MgSO_4$, 20 mM HEPES, and 5.5 mM glucose, pH 7.4), supplemented with 1 mM $Ca^{2+}$. When EGTA treatment was performed, aequorin measurements were recorded in KRB buffer plus 100 µM EGTA. Stimulation with EGF was performed as specified in the Figure legends. The experiments were terminated by lysing the cells with 0.01% Triton in a hypotonic $Ca^{2+}$-rich solution (10 mM $CaCl_2$ in $H_2O$), thus discharging the remaining aequorin pool. The light signal was collected and calibrated into $[Ca^{2+}]$ values, as previously described[20]. Extent of $Ca^{2+}$ waves was expressed as area under the curve (AUC) and maximal value of peak in box plot graphs (whiskers min to max). Statistical analysis was performed using GraphPad Prism.

**GCaMP measurements.** Measurements were performed using two different fluorescent $Ca^{2+}$-probe with different affinity for $Ca^{2+}$: GCaMP6f and GCaMP6m[13,39,70]. GCaMPs were developed for imaging rapid $Ca^{2+}$ peaks, like the one observed at ER-PM NCE contact sites. To localize the probe to the PM, where NCE contact sites are formed, we added the PM-targeting sequence of SNAP25, as used for Aequorin[5]. GCaMP-stable HeLa clones (PM-GCaMP6f and mitoGCaMP6m) were grown on 35-mm coverslips or MatTek 48 h prior to the acquisition of images. PM-GCaMP6f cells were subjected to 0.5 µg/ml overnight doxycycline induction the day before the recording of the experiment. For image acquisition, cells were washed and kept in KRB buffer supplemented with 1 mM $Ca^{2+}$ and glucose. To determine the PM $Ca^{2+}$ response, cells were placed on a 37 °C thermostat-controlled stage and exposed to 490/406 nm wavelength light using the Olympus fluorescent microscopy system equipped with a 20x oil objective acquiring 5fps for a total of 6 min. After recording baseline ratio, cells were stimulated with EGF under the indicated conditions. To assess the PM $Ca^{2+}$ response after HGF stimulation, HeLa cells were rinsed once with KRB buffer and allowed to adapt for approximately 5 min in KRB buffer supplemented with 1% serum to activate HGF. Similarly, for EGF experiments, the cells were placed on a 37 °C thermostat-controlled stage and exposed to light of 490/406 nm wavelength. After measuring the baseline ratio, the cells were stimulated with 250 ng/ml HGF, which was diluted in PBS-0.1%BSA. This same PBS-0.1%BSA solution was used as the control in the absence of HGF (-HGF condition). Fluorescent data collected were expressed as emission ratios. The increase in intensity was calculated with ImageJ using an ad hoc designed macro and the extent of $Ca^{2+}$ peaks was expressed as AUC and maximal value of peak in box plot graphs (whiskers min to max). Statistical analysis was performed using GraphPad Prism.

## Measurements of mitochondrial membrane potential with TMRM dye

Cells were grown on 35-mm round glass MatTek at 50% confluence the day before the experiment and then incubated with 1 nM TMRM dye (Catalog Number I34361, T668 Invitrogen) diluted in DMEM without FBS and red phenol supplemented with 2 mM L-glutamine and 25 µM verapamil (V4629, CAS 152-11-4, Sigma Aldrich) for 1 h at 37 °C and 5% $CO_2$. Then, the variation in the intensity of the dye was monitored by time-lapse microscopy. A Leica TCS SP8 AOBS inverted microscope with ×20 oil objective was used to take pictures every 2 s over a 5 min period. The assay was performed using an environmental microscope incubator set to 37 °C and 5% $CO_2$ perfusion. After cell incubation with the dye, EGF at the indicated doses was added and maintained in the media for the total duration of the time-lapse experiment. To evaluate changes in TMRM levels following HGF stimulation, cells were incubated with a 10 nM diluted solution of the dye in DMEM without red phenol, supplemented with 1% serum, 2 mM L-glutamine, and 25 µM verapamil. HGF, pre-activated in the same solution used for cell incubation, was then introduced to reach a final concentration of 100 ng/ml and remained in the media throughout the entire time-lapse experiment. The same DMEM without red phenol supplemented with 1% serum, 2 mM L-glutamine, and 25 µM verapamil solution was used as the mock control in the absence of HGF (-HGF condition). The increase in intensity was calculated with ImageJ using an ad hoc designed macro and the extent of the difference in mitochondrial potential was expressed as AUC and maximal value of peak in box plot graphs (whiskers min to max). Statistical analysis was performed using GraphPad Prism.

## Measurements of PM-localized ATP production with SNAP25-Luciferase (PM-Luc)

The pLVX-SNAP25-Luciferase HeLa cells were grown on 13-mm-round glass coverslips at 50% confluence and then analyzed by an IVIS luminometer. The system is composed of an upper part containing a highly sensitive photomultiplier arranged in a metal dark box to protect it and the temperature-controlled chamber from light exposure. The chamber (2 mm height and 265 µl volume) houses the cell sample seeded on a 13 mm-diameter coverslip during ATP measurement which was placed close to the photomultiplier. Cells were continuously perfused with temperature-controlled solutions in a water bath at 37 °C. A peristaltic pump allowed for liquid perfusion[73]. The three-phase acquisition started with 30 s of background recording in DMEM without red phenol (Lonza) supplemented with 2 mM L-glutamine. Then, the cell population was perfused with 25 µM beetle D-luciferin (Promega, E1601). Within 2–3 min, luciferase catalyzes light production, reaching a plateau after reacting with intracellular ATP. In the end, cells were perfused with EGF at the desired concentration and recorded for 2-3 min. The output signal was recorded and converted to kinetics via Hamamatsu Photonics software and the extent of the ATP was expressed as AUC in box plot graphs. Statistical analysis was performed using GraphPad Prism.

## Intracellular ATP levels detection

Intracellular ATP levels were assessed using the CellTiter-Glo kit (Promega Italia S.R.L.) following the manufacturer's protocol. Briefly, 5*10^3 HeLa cells were seeded in white-bottomed 96-well plates. The following day, cells were treated with the specified drugs (1 µM OMY and 20 µM Rotenone) for the indicated times. Subsequently, the cells were incubated with CellTiter-Glo Reagent, containing Luciferase substrate dissolved in a buffer, for 10 min at room temperature with gentle shaking, before luminescence measurements of ATP levels were taken using a GlowMax instrument. Luminescence data and statistical analyses were conducted using GraphPad Prism, with results presented as percentages relative to wild-type (WT) HeLa cells.

## CD147 internalization rescue experiment with histamine and succinate

Cells were subjected to MCU KD or mock transfection (see section "RNA interference"). After 4 days, control and MCU KD cells were seeded on 13-mm-round coverslips at 40-50% of confluency and CD147 internalization assay was performed (see section "CD147 internalization assay"). After the release of CD147 from 4 °C to 37 °C, cells were incubated with 10 µM histamine (Sigma) or pre-treated for 5 min and then incubated with 5 mM succinate (Sigma), or both in combination, either with or without high dose EGF. After internalization, cells were acid wash-treated and then fixed and processed for IF. Images were obtained using a Leica TCS SP8 confocal microscope equipped with a 63× oil objective and processed using ImageJ using an ad hoc designed macro (see section "CD147 internalization assay")[5].

## Cortical actin polymerization with MPAct/CaaX

Cells were seeded in 10 mm plates at 60% of confluence in a complete medium and then co-transfected with MPAct mCherry and YFP-CaaX vectors (selected plasmid constructs, corresponding sequence information is available on Addgene)[36]. After 36 h, transfected cells were seeded on 13 mm-round coverslips at a confluence of 40-50%. Cells were then stimulated with the indicated doses of EGF at 37 °C for 5 min. As a control, we also kept cells at 37 °C in the absence of the ligand, to set the basal levels of F-actin polymerization proximal to the PM. After internalization, cells were fixed and processed for IF. Images were obtained using a Leica TCS SP5 or TCS SP8 confocal microscope equipped with a 63× oil objective and processed using ImageJ. The membrane proximal actin (MPAct) value was generated by subtracting the background to each channel and then calculating the ratio of the MPAct/CaaX channels using an ad hoc designed macro. Ratiometric MPAct over CaaX intensities (here termed MPAct/CaaX) was used to measure relative local spatial heterogeneities in MPA density along the cell surface. Statistical analysis was performed using GraphPad Prism.

## Wound healing assay

HaCaT cells were seeded in 24-well plates equipped with proprietary treated plastic inserts ($2.5 \times 10^5$ cells per well) in complete medium and cultured until a uniform monolayer was formed. The inserts create a wound field with a defined gap of 0.9 mm for measuring the migratory rates of cells. The cell monolayer was scratched by carefully removing the plastic inserts and washed twice with 1X PBS to remove floating cells and to create a cell-free wound area. The closure of the wound was monitored by time-lapse video microscopy. A Nikon Eclipse Ti inverted microscope with 10× N2 objective was used to take pictures every 5 min over a 24 h period (as indicated in the figure legends). The assay was performed using an environmental microscope incubator set to 37 °C and constant 5% $CO_2$ perfusion. After the scratch, EGF at the indicated doses was maintained in the serum-starved media for the total duration of the time-lapse experiment. The wound front speed was calculated using Image J software. For the assay performed on interfered cells, cells were interfered following the same conditions already described in the 'RNA interference' section, plated the day before the experiment and stimulated in serum-starved medium supplemented with EGF at the indicated concentration.

**Wound healing assays upon Gefitinib treatment**. Wound healing assays were conducted to evaluate the effect of Gefitinib treatment on HaCaT cells. HaCaT cells were seeded at a density of $9 \times 10^5$ cells per well in 6-well plates with complete medium and cultured until a uniform monolayer was formed. On the day of the experiment, cells were serum-starved for 2 h prior to creating a wound field to measure cell migration. Subsequently, the cells were either stimulated with high EGF (100 ng/ml) or left unstimulated. At progressive time points (0, 12, and 18 h), either 5 μM Gefitinib or an equivalent volume of DMSO vehicle was added to the HaCaT cells, and cell migration was monitored for 48 hours post-stimulation. Images at each time point were captured using an EVOS microscope for each experimental condition. Migration rates of HaCaT cells were quantified by measuring the area of the wound gap using ImageJ software. The average of multiple measurements for each condition was used to calculate the migration rate, expressed as the ratio between the time points considered and the initial time. Statistical analyses were performed using GraphPad Prism.

## TPEF microscopy

For TPEF microscopy, cells were cultured on $22 \times 22 \times 0.17$ mm quartz slides (UQG Optics, United Kingdom). Three biological replicates were produced per each condition (-EGF, Control/High EGF, and RTN3 KD/High EGF. Cells were fixed in paraformaldehyde 4% and stored at 4 °C.

Prior to measurements, each quartz slide was mounted upside-down and sealed with a second 25x50x0.17 mm quartz slide (UQG Optics, United Kingdom), with cells placed in-between the two slides. We employed a home-built multimodal optical microscope featuring epi-detected Two-Photon Excited Fluorescence (TPEF), along with linear transmission light. This setup has been described in detail elsewhere[24,74]. Briefly, a compact Erbium-fiber multi-branch laser source (FemtoFiber Pro, Toptica Photonics, Germany) delivers 1560 nm pulses at a 40 MHz repetition rate with <100 fs duration. One branch is frequency-doubled to deliver narrowband 780-nm pump pulses, whereas a second branch (Stokes beam) is spectrally broadened and then frequency-doubled to allow the generation of tunable picosecond pulses in the 950-1050 nm range. Both the resulting beams feature a ≈ 1-picosecond duration. The average laser power at the sample plane is kept constant at 7.5 mW for the pump, and 0.5 mW for the Stokes, throughout the study, which limits cell photodamage ensuring non-invasiveness. The pump beam is used to excite TPEF interactions, whereas the red-shifted low-power Stokes beam at 1003 nm is used to obtain transmission images of the samples. The inverted-configuration microscopy unit includes high numerical aperture (NA) objectives that ensure subcellular spatial resolution ( ~ 280 nm lateral resolution, ~300 nm axial resolution, as for the Rayleigh criterion applied to nonlinear optical imaging): a water-immersion ×100 1.25NA 0.25 mm working distance (WD) illumination objective (C-Apochromat, Carl Zeiss, Germany) and an oil-immersion ×40 1.35NA 0.19 mm WD (CFI Super Fluor, Nikon, Japan) collection objective. The x-y-z motorized sample stage (Standa, Lithuania, and Mad City Labs Inc, U.S.A.) can scan large areas (up to $50 \times 50$ mm$^2$) while keeping the sample in focus. A photo-multiplier tube (PMT) (Hamamatsu Photonics, Japan) detects TPEF in the range 400-600 nm through optical filtering (FES0600, ThorLabs, Germany) at its inlet. The transmitted Stokes intensity is detected through a photodiode to obtain transmission images. All cell images in the present study have a dimension of 70 × 70 μm$^2$, 200 x 200 pixels (i.e., a pixel size of 350 × 350 nm). A 1.5-ms pixel dwell time is employed for image acquisition.

## TPEF image analysis

Images were processed via the Fiji-ImageJ software for image analysis. Both dark and bright 1-pixel outliers were median-filtered in order to correct extreme pixel values given by cosmic rays. An automated circular shift of image columns is used to compensate for distortion effects due to the serpentine-like motion of the motorized sample stage. The linear transmission of the Stokes is used to differentiate cells from the substrate and outline the cell area since cells appear as relatively dark regions due to a drop in laser transmission. As for mitochondrial NAD(P)H and FAD distribution, it was quantified from raw TPEF images acquired with constant integration time and laser power. A universal threshold of 0.75 a.u. was set to distinguish the endogenous signal from the diffused background that is presented outside of cells. The calculated percentage index of TPEF distribution inside cells is obtained as the ratio of TPEF area [μm$^2$] after signal thresholding over cell area [μm$^2$]. TPEF images are here displayed collectively normalized (TPEF signal range: 0–1).

## Statistical analysis

All statistical analyses were performed using GraphPad Prism. A two-sided Student's $t$ test was employed to obtain the statistical significance of differences between cells or endocytic structures. Non-parametric $U$-Mann–Whitney test was used to evaluate the statistical significance of TPEF distribution in cell populations. Exact p-values are shown in all panels. In the box plots, the lower and upper boundaries of the box are the first and third quartiles, with the median annotated with a line inside the box. The whiskers extend to the maximum and minimum values.

**Reporting summary**

Further information on research design is available in the Nature Portfolio Reporting Summary linked to this article.

## Data availability

All data generated in this study are available in the paper, in Supplementary Information or in Source Data file. Source data are provided with this paper.

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

## Acknowledgements

We thank P. De Camilli for HRP-KDEL cDNA, P. Tobias Meyer for cDNA of MPAct-mCherry, CFP-CaaX, and YFP-CaaX. We thank Rosalind Gunby for critically editing the manuscript. We thank the ALEMBIC facility at San Raffaele Scientific Institute, Milan, Italy for support in EM and CLEM analysis, in particular Eugenia Cammarota and Valeria Berno. We thank IEO Technological Units, in particular, Imaging and Flow Cytometry Units, Biochemistry and Structural Biology Unit and Cell Culture Unit. We thank Mario Romolo Faretta from IEO Imaging Development Unit for support with imaging experiments. This work was supported by grants from the European Research Council (ERC-CoG2020 101002280 to SS; ERC-Synergy 101071470-SHAPINCELLFATE to G.S.); Associazione Italiana per la Ricerca sul Cancro grant (AIRC IG 24415 to S.S.; AIRC IG 18621 and 5XMille 22759 to G.S.; AIRC IG 18988, AIRC IG 23060 and MCO 10000 to P.P.D.F.; AIRC IG-23670 to P.P.; AIRC IG 22811 to L.L.); World-wide Cancer Research grant (20-0094 to S.S.); Fondazione Piemontese per la Ricerca sul Cancro (L.L.); The Italian Ministry of University and Scientific Research grant (PRIN 2017 Prot. 2017E5L5P3 to S.S., C.T., and P.P.); The Italian Ministry of University and Scientific Research grant (PRIN 2020 Prot. 2020R2BP2E to PPDF, L.L.); The Italian Ministry of University and Scientific Research grant (PRIN 2020 Prot. 2020RRJP5L_005 to C.T.); The University of Milan grant (PSR2019 to S.S.); The Italian Ministry of Health (grant RF-2016-02361540 to P.P.D.F.); A-ROSE funding (P.P.); local funds from the University of Ferrara (P.P. and M.B.); FPRC 5xmille Ministero Salute (2017 PTCRC-INTRA 2020, project SEE-HER to L.L.); Associazione Italiana per la Ricerca sul Cancro AIRC fellowship (E.B., G.J., and G.M.); European Union project CRIMSON (under Grant Agreement No. 101016923 to D.P.); and The Italian Ministry of Health (Ricerca Corrente 2023-2024).

## Author contributions

Conceptualization: S.S., P.P.D.F., M.B., P.P., G.S., and C.T.; Methodology: S.S., P.P.D.F., M.B., P.P., A.R., C.T., M.Q., M.G.M., G.S., H.A., D.P., and L.L.; Investigation: D.M., E.B., A.R., S.F., G.M., G.J., G.C., M.Q., I.S.L., H.A., A.B., F.M., F.V., and L.L.; Visualization: D.M., S.S., A.R., S.F., M.G.M., and A.B.; Funding acquisition: S.S., P.P.D.F., P.P., C.T., D.P., G.S., and L.L.; Supervision: S.S. and P.P.D.F.; Writing – original draft: S.S. and P.P.D.F.; Writing – review & editing: S.S., P.P.D.F., D.M., M.B., G.S., P.P., and M.G.M.

## Competing interests

The authors declare no competing interests.
