## [Peer Review File · Nature Communications]

REVIEWER COMMENTS

Reviewer #1 (Remarks to the Author):

Mechanisms of clathrin-independent endocytosis are under active scrutiny. In the current manuscript, the authors have investigated the tripartite link between tubular plasma membrane invaginations that are generated by a clathrin-independent process, membranes of the endoplasmic reticulum decorated by reticulon-3, and mitochondria. A functional model is presented that links calcium homeostasis to membrane scission for non-clathrin endocytic uptake into cells.

With their work, the authors address a highly relevant research question with an exceptional density of data, including sophisticated tomography and correlated light and electron microscopy. This study is important for several reasons: Linking the processing of tubular plasma membrane invaginations to calcium homeostasis on ER and mitochondrial membranes is highly innovative, and established in the current manuscript based on thorough experimentation. Furthermore, linking the reticulon-3 mechanism that Sigismund and Di Fiore have pioneered to other growth factor signaling (here: HGFR/MET) is a great step forward in the exploration of clathrin-independent endocytosis.

There is only one point that I would bring to the authors attention for possible optimization. On line 265-266, it is concluded that “actin and dynamin cooperate in NCE TI fission”. In the NCE of Shiga toxin, it had previously been argued that dynein motor activity and the BAR domain protein endophilin were involved in scission of tubular endocytic pits, in addition to actin and dynamin (DOI: 10.1038/nature14064). Does interference with dynein also affect the uptake process that is studied in the current manuscript?

Reviewer #2 (Remarks to the Author):

This is an interesting study which builds upon previous investigations by this group linking the internalisation of the EGFR via NCE with the formation of PM-ER and now mitochondria MCSs. While the results are interesting and constitute a nice hypothesis, I am of the opinion that they are too preliminary at this point to be considered as a complete study. Notably, this investigation has not elucidated the molecular players linking ER and mitochondria that are involved in this process. Candidates to play this role would presumably include MAM-localised proteins already known to be required for ER-mitochondria contacts. Elucidating the specific players involved and devising ways

to block this interaction in order to infer effects in EGFR NCE and/or downstream signalling would strengthen notably the paper. There is another point that conceptually is a bit difficult to understand at this point. The authors state that these contacts are formed specifically from NCE invaginations. However, they also suggest that these contacts provide the necessary energy to form these invaginations. It is therefore important that the authors clarify the order of events that they expect here. The use of RTN3 KD to unravel this is slightly confusing as this factor has already been shown by the authors to be involved in the formation PM-ER contacts - it would be important to use a mutant EGFR that cannot be recruited to NCE invaginations or another independent means to block NCE endocytosis to clarify if the PM-ER-mitochondrial contacts occur from active EGFR at the PM prior to their NCE engagement or not. This point is important as one would expect large consequences for the control of cell behaviour if EGFR signalling impacted mitochondria in this way irrespective of the coupling of this process with NCE or not. In terms of discounting CME as involved, experiments involving EGFR mutants unable to bind to AP2 would be more convincing than simply knocking down AP2. Also, one would expect that blocking CME would in turn increase NCE and therefore PM-ER-mitochondrial contacts (if this is what the authors are inferring, ie NCE formation increases these contacts, which is not clear yet at this point), so it is not obvious why manipulating AP2 is without effect. Other points include 1) whether the authors have investigated if these contacts are really exclusively from the PM or can also involve endosomal EGFR, as there are numerous previous reports describing the capacity for EGFR-positive endosomes to engage in multiple types of MCSs; 2) have the authors checked if there are any changes in mitochondrial fusion/fission as it is now established that MCS formation is tightly linked with the control of mitochondrial morphology, which has deep effects on its function; 3) currently not very sure how is the formation of the PM-ER-mitochondrial contact linked to the changes in cell migration and actin cytoskeleton. Do changes in mitochondrial engagement and function affect EGFR downstream signalling? The authors show that in fact blocking ATP formation with oligomycin does not seem to have any effect in this, so how can the engagement of mitochondria be linked to the EGFR-dependent actin remodelling effects?

Minor comments: 1) not sure why oligomycin is not affecting cytosolic ATP levels, should the authors try a different concentration as normally oligomycin should reduce these; 2) Fig 5c is not discussed in the results so not sure what is the relevance of this.

Reviewer #3 (Remarks to the Author):

The epidermal growth factor receptor (EGFR) is a receptor tyrosine kinase that regulates many aspects of cell physiology. Upon binding ligand, EGFR triggers activation of many signaling pathways and concomitantly undergoes endocytosis. While EGFR undergoes clathrin-mediated endocytosis upon stimulation with lower concentrations of ligand (EGF), at higher doses of EGF, the receptor undergoes non-clathrin endocytosis (NCE) in some cell types. The Sigismund group has led in the development of our understanding of the molecular mechanism of NCE of EGFR. Specifically, previous work showed that at high doses of EGF, there is a formation of contact sites of

endoplasmic reticulum and plasma membrane requiring ER resident protein reticulon-3 (RTN3). NCE of EGFR in this circumstance requires RTN3 and Ca²⁺ release by IP3R from the ER. This current manuscript examines the outcome and function of NCE of EGFR.

Using CD147 as a marker of NCE (alongside EGFR), EM tomography shows formation of tri-partite contact sites of ER, PM and mitochondria upon treatment of high doses of EGF. The formation of ER-mitochondria contact sites is also observed by confocal microscopy upon treatment with high dose EGF, which requires RTN3. Then, intracellular Ca²⁺ signaling was measured using GCaMP6f, which can be targeted to specific organelle membranes. GCaMP6f targeted to the plasma membrane revealed Ca²⁺ oscillations that were dependent on RTN3 and IP3R, as well as on mitochondrial calcium transporters. Similar Ca²⁺ oscillations were observed when targeting GCaMP6f to the mitochondrial matrix. High dose EGF enhanced mitochondrial metabolism, observed by an increased fluorescence of the membrane potential sensitive dye TMRM, increased two photon excitation to detection of NADH, and plasma membrane targeted, luciferase-dependent detection of ATP levels. The enhancement of each of these parameters upon high dose EGF treatment was abolished by perturbation of RTN3, IP3R, and mitochondrial metabolic functions. Interestingly, similar results were observed with MET stimulated at high doses of its ligand HGF.

Next, the contribution of mitochondrial function to NCE was assessed. Perturbation of mitochondrial Ca²⁺ uptake or ATP production resulted in reduction of EGFR NCE and elongation of plasma membrane tubular invaginations, consistent with impairment of vesicle scission from the plasma membrane. The NCE of CD147 in cells depleted of mitochondrial Ca²⁺ buffering activity could be rescued by treatment with histamine, which induces Ca²⁺ oscillations in a different manner, or by treatment with succinate, which enhances mitochondrial ATP production independently of Ca²⁺. High dose EGF also induced cortical actin polymerization, as observed by Arp2/3 localization as well as the probe MPAct, each dependent on RTN3, IP3R and mitochondrial functions. Lastly, perturbation of RTN3, mitochondria Ca²⁺ buffering, mitochondrial ATP production, and actin polymerization all impaired high dose EGF-stimulated wound healing velocity in HaCaT cells.

Overall, this is a very interesting study that links EGFR signaling to induction of a mechanism for localized ATP production by mitochondria in close proximity to the sites of non-clathrin endocytosis. The evidence to support the major conclusions is largely very strong, with multiple assays to measure each phenomenon and multiple perturbation strategies to support the roles of specific proteins/molecules or processes. The expansion of NCE to also regulate the endocytosis of MET is also interesting, as this is quite novel and shows for the first time that the impact of this NCE pathway extends beyond EGFR. Furthermore, the linking of EGFR NCE to cell migration can provide insightful cell physiological impact of the NCE pathway. As such, this impactful study will be of interest to a broad audience of cell biology researchers. A few comments should be addressed to strengthen some of the conclusions, as outlined below

Specific comments

1) The formation of tripartite membrane contact sites following EGF stimulation is proposed, based on the proximity of plasma membrane tubular invaginations, ER and mitochondria observed in EM tomograms and by confocal microscopy. The RTN3-dependence of the membrane contact sites at the cell periphery induced by high dose EGF is potentially interesting. However, signal overlap by confocal microscopy is not sufficient to resolve membrane contact sites, which require proximity of <30 nm of the two compartments. Can formation of ER and mitochondria contact sites be observed by other methods such as proximity ligation (PLA)? The study does include observation of membrane contact sites by EM tomography, but the resolving the effects of perturbation of RTN3 on ER-mitochondria contact sites is important, and this appears to only have been examined by confocal microscopy.

Also, what is the significance of the lack of effect of RTN4 on putative membrane contact sites formed upon treatment with high dose EGF? It is not clear if RTN4 was previously shown to be involved in ER membrane contact sites in other contexts, or if RTN4 has other functions in the ER unrelated to the membrane contacts suggested to form following high dose EGF. It would also be informative to consider the effect of perturbation of other well established ER membrane contact site proteins such as VAPA/B on the formation of these tripartite membrane contact sites between the PM, ER, and mitochondria. This would allow some context to these contact sites – are they similar to contacts already described or a different type of ER-dependent membrane contact site?

2) The experiments shown in Figure 7 examine how NCE of EGFR may control cell migration. This is potentially interesting. However, the effects of high dose EGF on regulation of cell migration are somewhat perplexing, since high dose EGF leads to enhanced degradation of EGFR – around 60% of EGFR is degraded upon high dose EGF within 1 h, based on some prior work by these authors. Since these cell migration experiments rely on long-term stimulation with high dose EGF (12-24h, e.g. figure 7E), it is not clear how EGFR-derived signals can regulate this cell migration phenomenon beyond 1-2 h of initial stimulation with high dose EGF. It is thus sufficient to have a smaller population of EGFR continue to undergo NCE beyond the initial 1h period? Some suggestions to address this would be to show the levels of EGFR and key signals such as pEGFR in these cells at various timepoints following high dose EGFR stimulation. It may also be useful to consider adding EGFR TKI such as erlotinib after ~12 h of the high dose EGF stimulation, which would be expected to suppress the enhanced cell migration if this still depends on sustained NCE-dependent signaling from a smaller population of EGFR that has not been subject to degradation.

3) The experiments that link mitochondrial ATP production to EGFR NCE, regulation of cortical actin, and cell migration suffer at least in part from the potentially broad effects of perturbation of

mitochondrial activity. This applies to disruption of mitochondrial activity with oligomycin (OMY), as well as to disruption of mitochondrial buffering capacity. It is very difficult to resolve what the effects of various perturbations may be on ATP levels and mitochondrial activity in the absence of high dose EGF. This is in part due to the apparent normalization of ATP measurements (e.g. Figure S3G) to the baseline within each condition, rather to a single standard condition (e.g. control). As a result, it would appear that the results shown linking mitochondrial ATP production to EGFR NCE may show a higher sensitivity of EGFR NCE to mitochondrial activity and ATP levels, but do not yet show a selective role of the EGF-stimulated enhanced ATP production. Since addressing this experimentally will be very difficult, some tempering of the conclusions about the functional role of EGF-stimulated enhancement of mitochondrial ATP production should be considered. This does not detract from the high level of enthusiasm of this reviewer for the study.

REVIEWER COMMENTS

Reviewer #1

There is only one point that I would bring to the authors attention for possible optimization. On line 265-266, it is concluded that “actin and dynamin cooperate in NCE TI fission”. In the NCE of Shiga toxin, it had previously been argued that dynein motor activity and the BAR domain protein endophilin were involved in scission of tubular endocytic pits, in addition to actin and dynamin (DOI: 10.1038/nature14064). Does interference with dynein also affect the uptake process that is studied in the current manuscript?

R: The point raised by the reviewer is an interesting one. Concerning Endophilin-A1/2, we have shown that they are not involved in NCE (Caldieri et al., 2017), although we cannot exclude a role for other BAR domain proteins in the process. In addition, following the reviewer’s suggestion we have performed experiments to address the role of dynein in NCE. When we used the dynein inhibitor, ciliobrevin D, we found that it inhibits CD147/EGF NCE, but not TfCME (**Fig. S7A,B**). Dynein seems to act at the initial steps of NCE as it reduces ER-PM contact site formation (**Fig. S7C**) and TIs formation, but not CCPs (**Fig. S7D**). Thus, it does not phenocopy Ca^{2+} inhibition, which instead affects only the last step of TI fission. The involvement of dynein in the early steps of NCE might be linked to its role in cortical ER motility (Woźniak M.J., JCS 2009). Our data do not exclude, however, an involvement of dynein also at the fission step in concert with actin and dynamin, as previously shown for the NCE of Shiga toxin.

Reviewer #2:

General points

...Notably, this investigation has not elucidated the molecular players linking ER and mitochondria that are involved in this process. Candidates to play this role would presumably include MAM-localised proteins already known to be required for ER-mitochondria contacts. Elucidating the specific players involved and devising ways to block this interaction in order to infer effects in EGFR NCE and/or downstream signalling would strengthen notably the paper

R: In this study we report for the first time the formation of a tripartite organelle platform (involving PM, ER and mitochondria) at sites of EGFR activation at the PM, and we provide evidence for a functional crosstalk between EGFR signaling and mitochondrial energetics, capable of integrating both attenuator and effector function of EGFR signaling. We also describe RTN3 as a critical player mechanistically involved in the formation of this platform, at variance with its related family member RTN4. Notably, in our previous work (Caldieri et al., 2017), we have shown that RTN3 is able to FRET with active EGFR upon high dose of EGF, suggesting that an EGFR/RTN3 complex (direct or indirect) might be involved in the tethering process. We acknowledge that defining the precise molecular mechanisms of membrane tethering is a relevant question, it should be appreciated, however, that this will require a substantial experimental effort, including: i) defining whether the interaction between EGFR and RTN3 is direct or indirect, ii) if indirect, identifying the intervening proteins, iii) if direct, mapping the surface of EGFR/RTN3 interaction; iv) generating mutants and performing reconstitution assays in a the appropriate KD background; v) elucidating the involvement of other accessory tethering factors in contact formation. This defines a research project *per se*, that we are actively pursuing, and on which we will report in due time.

There is another point that conceptually is a bit difficult to understand at this point. The authors state that these contacts are formed specifically from NCE invaginations. However, they also suggest that these contacts provide the necessary energy to form these invaginations.

R: Agree. What we reported in the original manuscript is that the energy provided by mitochondrial ATP is required for the last step of membrane fission. Indeed, the inhibition of ATP caused the elongation of tubular invaginations (TIs) and the inhibition of their fission, thereby phenocopying the KD of dynamin (**Fig. 5D**). Prompted by the question of this reviewer, we have now investigated whether mitochondrial ATP is also needed for membrane invagination or contact site formation. To do so, we performed an extensive EM analysis in which we measured the number of NCE-TIs and of ER-PM contact sites upon oligomycin treatment. We found that oligomycin causes a reduction in the number of TIs, suggesting that ATP is also needed at the PM invagination step (**new Fig. S5D**). The ER-PM contacts are instead not affected (**new Fig. S5C**), showing that their formation does not require mitochondrial ATP (see also next point).

It is therefore important that the authors clarify the order of events that they expect here. The use of RTN3 KD to unravel this is slightly confusing as this factor has already been shown by the authors to be involved in the formation PM-ER contacts - it would be important to use a mutant EGFR that cannot be recruited to NCE invaginations or another independent means to block NCE endocytosis to clarify if the PM-ER-mitochondrial contacts occur from active EGFR at the PM prior to their NCE engagement or not. This point is important as one would expect large consequences for the control of cell behaviour if EGFR signalling impacted mitochondria in this way irrespective of the coupling of this process with NCE or not.

R: Agree. This is an important point. We have investigated by EM analysis whether the ER-mitochondrial recruitment occurs before or after the formation of PM invaginations. Our data show that the tripartite contact sites are triggered by EGF (or HGF) before the invagination is formed (**new Fig. 1A and Fig. 4B**). This confirms that receptor activation triggers the formation PM-ER-mitochondria contacts, that are required for the formation of tubular invaginations. In agreement, we have previously shown that the abrogation of contact sites by RTN3 KD affects PM tubular invagination (Caldieri et al., Science, 2017). Mitochondrial ATP - generated thanks to the ER-mitochondrial interplay - is needed both for TIs formation and, at a later stage, for the final step of TI fission.

In terms of discounting CME as involved, experiments involving EGFR mutants unable to bind to AP2 would be more convincing that simply knocking down AP2. Also, one would expect that blocking CME would in turn increase NCE and therefore PM-ER-mitochondrial contacts (if this is what the authors are inferring, ie NCE formation increases these contacts, which is not clear yet at this point), so it is not obvious why manipulating AP2 is without effect.

R: In reference to this question, we note that EGFR-CME and -NCE mechanisms are independent of each other. Both EGFR and CD147 are internalized via NCE upon EGF in unperturbed cells. When CME is inhibited, there is no alteration in the recruitment of these cargoes to NCE-TIs (Caldieri et al., 2017). Similarly, the inhibition of NCE does not alter the recruitment of EGFR to CCPs nor its internalization via CME (Caldieri et al., 2017). Thus, differently from other clathrin-independent mechanisms (e.g., CLIC-GEEC), EGFR-NCE is not a compensatory mechanism that gets overactivated when clathrin is inhibited. Given this premises, we believe that AP2 KD represents a suitable control to probe into the specificity of NCE in the regulation of a given phenotype.

Other points include:

1) whether the authors have investigated if these contacts are really exclusively from the PM or can also involve endosomal EGFR, as there are numerous previous reports describing the capacity for EGFR-positive endosomes to engage in multiple types of MCSs...

R: The current study has raised a number of new questions that need to be addressed in the future. ER contacts with EGFR-positive endosome/MVBs have been documented (Rowland et al, Cell et al, 2014; Wu, Dev Cell, 2021; Eden et al., 2016; White et al., 2006). Whether these are the same type of contacts that we observed, remains to be established. As we discussed in our reply to first general point, we are performing a screening of known tethering factors to understand whether the contacts herein described employ the same basic machinery that is involved in other contact sites. Related to this, we note that the long isoform of RTN3 has been implicated in the tethering of ER tubules and EGFR-positive endosomes (Wu et al., Dev Cell, 2021). Interestingly, we have previously shown that the short isoform of RTN3 is sufficient to rescue the NCE internalization defect upon RTN3 KD (Caldieri et al., 2017). This raises the intriguing possibility that specific RTN3 isoforms are involved in different types of ER contacts, possibly interacting with specific sets of tethering factors, an issue that we are currently investigating. We have included more discussion on this specific point (pages 11-12 of the revised manuscript).

2) have the authors checked if there are any changes in mitochondrial fusion/fission as it is now established that MCS formation is tightly linked with the control of mitochondrial morphology, which has deep effects on its function;

R: Agree. We checked this point and we found that, at early time points of EGF stimulation (5 minutes), when the tripartite organelle platform is formed and active in endocytosis, no direct effect on mitochondrial morphology is observed (new **Fig. S1D**).

3) currently not very sure how is the formation of the PM-ER-mitochondrial contact linked to the changes in cell migration and actin cytoskeleton. Do changes in mitochondrial engagement and function affect EGFR downstream signalling? The authors show that in fact blocking ATP formation with oligomycin does not seem to have any effect in this, so how can the engagement of mitochondria be linked to the EGFR-dependent actin remodelling effects?

R: We apologize for the lack of clarity. The experiments with oligomycin in **Figs. 3H-5B-5D-6G and S4E-F** are performed acutely (15 min in total) and at a low dose (1 μ M), in order not to affect the total ATP pool nor EGFR activation (as shown experimentally in **Fig. S4E and new Fig. S4F**), but to selectively and transiently impact on the release of ATP at the PM induced by EGF, as confirmed by the PM-luciferase assay (**Fig. 3H**). Thus, the effects of short-term oligomycin treatment are not due to a general impairment of the total ATP pool. This was instrumental to show that EGF induced a local release of ATP by PM-localized mitochondria, which is crucial for cortical actin polymerization and for the fission of tubular invagination.

Minor comments:

1) not sure why oligomycin is not affecting cytosolic ATP levels, should the authors try a different concentration as normally oligomycin should reduce these;

R: Agree. In the revised manuscript, we added data with long treatment with oligomycin (same time length as that used for rotenone). Under these conditions, there was a significant reduction in the total ATP levels comparable to that obtained in rotenone-treated cells (new **Fig. S4F**).

2) Fig 5c is not discussed in the results so not sure what is the relevance of this.

R. We apologize for this mistake. We have now included the description of this figure in the main text (page 8). This experiment was performed to corroborate the role of a local release of ATP in EGFR-NCE. Indeed, we selectively depleted the PM-localized ATP pool using

SNAP25-luciferase, not as an ATP sensor, but as an ATP-consuming enzyme. In presence of an excess of luciferin, CD147/EGF internalization was inhibited in cells expressing SNAP25-luciferase, but not in control cells (**Fig. 5C**, **Fig. S5A**). In contrast, Tf endocytosis remained unaffected (**Fig. S5B**), arguing that NCE specifically requires the PM-localized ATP pool for its execution, at variance with CME.

Reviewer #3

Specific comments

1) The formation of tripartite membrane contact sites following EGF stimulation is proposed, based on the proximity of plasma membrane tubular invaginations, ER and mitochondria observed in EM tomograms and by confocal microscopy. The RTN3-dependence of the membrane contact sites at the cell periphery induced by high dose EGF is potentially interesting. However, signal overlap by confocal microscopy is not sufficient to resolve membrane contact sites, which require proximity of <30 nm of the two compartments. Can formation of ER and mitochondria contact sites be observed by other methods such as proximity ligation (PLA)? The study does include observation of membrane contact sites by EM tomography, but the resolving the effects of perturbation of RTN3 on ER-mitochondria contact sites is important, and this appears to only have been examined by confocal microscopy.

R: We acknowledge the Reviewer's point. Within these tripartite structures, we encounter two distinct types of contact sites: the PM-ER and the ER-Mitochondria interfaces. Our prior EM analysis has shown that the depletion of RTN3 abolishes the formation of PM-ER contacts (Caldieri et al., 2017). Herein, we present evidence that ablation of RTN3 abolishes also the EGF-induced proximity between ER-Mitochondria and PM (**Fig. 1G**). Notably, although we acknowledge that these data were obtained at a lower resolution than EM, it is however at a higher resolution than confocal microscopy, as the staining was performed on ultrathin sections of 100 nm. By doing this, while the *x-y* resolution remains at the diffraction limit (200 nm), the axial resolution is improved of ~10-fold vs. standard confocal microscopy (Takizawa T. et al., 2015, PMID: 26884817). Indeed, the ER-Mitochondrial contacts visualized by this method were validated by CLEM demonstrating the resolution power of the methodology (**Fig. 1F**, **Fig. S1B**).

Our findings can be interpreted in two plausible scenarios: firstly, RTN3 may exclusively participate in PM-ER contacts, and their disruption consequently leads to the withdrawal of ER-Mitochondria from the PM; alternatively, RTN3 might directly engage in the formation of ER-Mitochondria contacts as well. Our present data do not allow to distinguish between these two hypotheses. We have discussed this point at pages 11-12 of the revised manuscript.

In compliance with the reviewer's suggestion, we performed PLA between the mitochondrial channel VDAC and the IP3R channel on the ER. Our analysis did not reveal an increase in ER-Mitochondria contact sites following high-dose EGF stimulation, and we observed no reduction of contacts upon RTN3 KD (refer to the figure below). This observation may imply that RTN3 does not play a major role in establishing ER-Mitochondria contacts induced upon EGF, but it is only involved in PM-ER contact formation. Nevertheless, it's worth noting that PLA lacks the spatial resolution required to exclusively examine the fraction of contacts in close proximity to the PM, as the PLA dots typically exceed 0.5-1 μm in diameter and their position is the result of their binding to a DNA probe (which might land quite far away from the original site). Consequently, we remain cautious in interpreting these data, as they do not definitively exclude the possibility of RTN3 involvement in the subset of ER-Mitochondrial contacts regulated by EGF in the proximity of the PM. For this reason, we did not include this

data in the manuscript; however, if the Reviewer thinks this is relevant, we can include it with some further discussion.

Figure Proximity Ligation Assay (PLA) of IP3R3-VDAC1 upon EGFR-NCE activation. PLA assay was performed as described (D'Eletto M. et al., Cell Reports, 2018). HeLa cells were subjected or not to RTN3 KD and stimulated or not with high EGF (100 ng/ml). Cells were subsequently fixed in 4% paraformaldehyde, permeabilized in 0.1% TX-100, and subjected to an antigen retrieval process in 6M Urea pH 9 at 80°C, as previously described (Hayashi T. et al., Histochem Cell Biol, 2011). Following blocking in 5% goat serum, 2% BSA, 0.1% TX-100 in PBS, cells were incubated with primary antibodies (anti-IP3R-3, BD #610312; anti-VDAC1-Porin, abcam, #ab15895) for 60 mins at 37°C. The proximity ligation assay (PLA) protocol was carried out using the Duolink II Orange in situ PLA kit (Merck Life Science S.R.L.) according to the manufacturer's instructions. Briefly, after primary antibody incubation, PLA probes (anti-Rabbit minus and anti-Mouse plus) were hybridized with the samples for 60 minutes at 37°C, followed by proximity ligation for 30 minutes at 37°C and amplification for 60 minutes at 37°C, allowing for the detection of PLA signals coming from the interaction between IP3R-3 and VDAC1. Left, representative PLA images of HeLa cells subjected to RTN3 KD or mock-control transfection, stimulated or not with high EGF, plus HeLa cells incubated with only one of the two primary antibodies as negative control to assess the specificity of PLA signals. Blue, DAPI. Bar, 10 μ m. Right, quantitative analysis of IP3R3-VDAC1 interactions reported as the mean values of PLA dots/cells \pm SD are reported as a percentage relative to unstimulated control. N, number of cells: Control -EGF N=155, Control +EGF N=196, RTN3 KD -EGF N=138, RTN3 KD +EGF N=189. The average of three independent experiments is shown (n=3). P-value (Each Pair Student's t-test, two-tailed): ns, not significant. Specificity of anti-IP3R and -VDAC1 antibodies in IF was validated by KD of the corresponding gene.

Also, what is the significance of the lack of effect of RTN4 on putative membrane contact sites formed upon treatment with high dose EGF? It is not clear if RTN4 was previously shown to be involved in ER membrane contact sites in other contexts, or if RTN4 has other functions in the ER unrelated to the membrane contacts suggested to form following high dose EGF.

R: We believe that the differential effect of RTN3-KD vs. RTN4-KD is relevant. RTN3 and RTN4 belong to the same family of proteins, localized to the cortical tubular ER thanks to their reticulon homology domain (RHD). RTN4 has been directly implicated in the shaping of ER tubules (Voeltz et al., 2006, Caldieri et al., 2017, Wu et al., 2021). Conversely, increasing evidence suggests that RTN3 might exert a more restricted function in specifying regions of the ER involved in membrane contact sites, without affecting ER morphology (Shi et al., 2014, Caldieri et al., 2017, Wu et al., 2021). Thus, the lack of effect of RTN4 provides a specificity control that the phenotypes observed upon RTN3-KD are not linked to general alterations in ER shaping but to the unique function of RTN3 in contact sites formation.

The molecular determinants (and the exact mechanism) underlying the different biological functions of RTN3 and RTN4 remain to be established and we have discussed some possibilities in our revised manuscript (pages 11-12).

It would also be informative to consider the effect of perturbation of other well established ER membrane contact site proteins such as VAPA/B on the formation of these tripartite membrane contact sites between the PM, ER, and mitochondria. This would allow some context to these contact sites – are they similar to contacts already described or a different type of ER-dependent membrane contact site?

R: This is an interesting point. We are presently performing a screening of known tethering factors in order to understand whether these contacts use the same basic machinery that has been described for other contact sites. We can anticipate that this seems to be the case, and we are deeply investigating the molecular mechanism. However, the work is still preliminary as we have only performed functional endocytic assays, and it will require further investigations to dissect the underlying molecular mechanism.

2) The experiments shown in Figure 7 examine how NCE of EGFR may control cell migration. This is potentially interesting. However, the effects of high dose EGF on regulation of cell migration are somewhat perplexing, since high dose EGF leads to enhanced degradation of EGFR – around 60% of EGFR is degraded upon high dose EGF within 1 h, based on some prior work by these authors. Since these cell migration experiments rely on long-term stimulation with high dose EGF (12-24h, e.g. figure 7E), it is not clear how EGFR-derived signals can regulate this cell migration phenomenon beyond 1-2 h of initial stimulation with high dose EGF. It is thus sufficient to have a smaller population of EGFR continue to undergo NCE beyond the initial 1h period? Some suggestions to address this would be to show the levels of EGFR and key signals such as pEGFR in these cells at various timepoints following high dose EGFR stimulation. It may also be useful to consider adding EGFR TKI such as erlotinib after ~12 h of the high dose EGF stimulation, which would be expected to suppress the enhanced cell migration if this still depends on sustained NCE-dependent signaling from a smaller population of EGFR that has not been subject to degradation.

R: We thank the Reviewer for highlighting this point and providing suggestions. Indeed, it is to understand whether the long-term effects on migration are dependent on a small population of EGFRs continuously internalizing via NCE.

To this aim we have performed the experiments suggested by the Reviewer:

i) we checked the level of phosphorylated EGFR and downstream signaling molecules upon prolonged EGF stimulation (*i.e.*, 6h, 12h, 18h), showing that indeed there is a residual fraction of active receptors even after chronic EGF exposure (new **Fig. S9A**);

ii) we repeated the wound healing assay upon inhibition of the EGFR kinase activity at different time points of EGF stimulation (0, 6, 12, 18 h) and we found that kinase inhibition affects cell migration even if applied at late time points after EGF addition (12h or 18h). These data suggest that a small fraction of active EGFRs recruited to the NCE platform might be responsible for sustaining the migratory phenotype, as pointed by the Reviewer (new **Fig. S9B**). Whether this effect is linked to the platform itself or to signaling emanating by NCE-derived vesicles/endosomes harboring the EGFR, remains to be established.

3) The experiments that link mitochondrial ATP production to EGFR NCE, regulation of cortical actin, and cell migration suffer at least in part from the potentially broad effects of perturbation of mitochondrial activity. This applies to disruption of mitochondrial activity with oligomycin (OMY), as well as to disruption of mitochondrial buffering capacity. It is very difficult to resolve what the effects of various perturbations may be on ATP levels and mitochondrial activity in the absence of high dose EGF. This is in part due to the apparent normalization of ATP measurements (e.g. Figure S3G) to the baseline within each condition, rather to a single standard condition (e.g. control). As a result, it would appear that the results shown linking mitochondrial ATP production to EGFR NCE may show a higher sensitivity of EGFR NCE to mitochondrial activity and ATP levels, but do not yet show a selective role of the EGF-stimulated enhanced ATP production. Since addressing this experimentally will be very difficult, some tempering of the conclusions about the functional role of EGF-stimulated enhancement of mitochondrial ATP production should be considered. This does not detract from the high level of enthusiasm of this reviewer for the study.

R: This is an important point and we agree with Reviewer's criticism. We toned down our conclusions discussing on the potentially broad effects of the perturbation of mitochondrial activity which might extend beyond EGFR endocytosis and signaling (page 12 of the revised manuscript).

REVIEWERS' COMMENTS

Reviewer #1 (Remarks to the Author):

The authors have responded with new experiments to the point that I had raised on the initial version of this manuscript. The observed results further reinforce this study, which clearly is of interest to a general readership in the life sciences.

Reviewer #2 (Remarks to the Author):

I appreciate the thorough review by the authors which I believe have improved the manuscript and added some clarity to some of the observations. I just have a remaining remark to make:

Regarding the first point in the rebuttal for reviewer 2, the authors argue that defining the PM-ER-mitochondria complex components is a separate project and they eloquently explain why this is not possible to be achieved in the present report. Specifically they note the following:

"We acknowledge that defining the precise molecular mechanisms of membrane tethering is a relevant question, it should be appreciated, however, that this will require a substantial experimental effort, including: i) defining whether the interaction between EGFR and RTN3 is direct or indirect, ii) if indirect, identifying the intervening proteins, iii) if direct, mapping the surface of EGFR/RTN3 interaction; iv) generating mutants and performing reconstitution assays in a the appropriate KD background; v) elucidating the involvement of other accessory tethering factors in contact formation."

Can this paragraph please be included in the discussion as it is written above so that it is clear that this part of the investigation is incomplete and requires further research? I think this will help in appreciating the current state of the investigation and where it should be going in the near future.

If this paragraph is added to the discussion, I would be happy to support publication.

Reviewer #3 (Remarks to the Author):

The revised manuscript addresses the points raised during review of the initial submission quite well. I appreciate the authors attempting the PLA experiment that was suggested, and based on the result of this experiment, I agree with the authors that this does not need to be included in the

manuscript at this time. Clearly, there is some interesting work that can be done in the future to probe membrane contact sites involved in this process further, but that such experiments are beyond the scope of the current study. I have no further comments. This is a very exciting study that will be of significant and broad interest to cell biologists.

REVIEWER COMMENTS

We thank all reviewers for their appreciative comments. Reviewers 1 and 3 had no further concerns. Below, our reply to Reviewer 2.

Reviewer #1

The authors have responded with new experiments to the point that I had raised on the initial version of this manuscript. The observed results further reinforce this study, which clearly is of interest to a general readership in the life sciences.

Reviewer #2

I appreciate the thorough review by the authors which I believe have improved the manuscript and added some clarity to some of the observations. I just have a remaining remark to make: Regarding the first point in the rebuttal for reviewer 2, the authors argue that defining the PM-ER-mitochondria complex components is a separate project and they eloquently explain why this is not possible to be achieved in the present report. Specifically they note the following: "We acknowledge that defining the precise molecular mechanisms of membrane tethering is a relevant question, it should be appreciated, however, that this will require a substantial experimental effort, including: i) defining whether the interaction between EGFR and RTN3 is direct or indirect, ii) if indirect, identifying the intervening proteins, iii) if direct, mapping the surface of EGFR/RTN3 interaction; iv) generating mutants and performing reconstitution assays in a the appropriate KD background; v) elucidating the involvement of other accessory tethering factors in contact formation."

Can this paragraph please be included in the discussion as it is written above so that it is clear that this part of the investigation is incomplete and requires further research? I think this will help in appreciating the current state of the investigation and where it should be going in the near future.

If this paragraph is added to the discussion, I would be happy to support publication.

R: The sentence suggested by the Reviewer has been included in the Discussion (page 12, line 20).

Reviewer #3

The revised manuscript addresses the points raised during review of the initial submission quite well. I appreciate the authors attempting the PLA experiment that was suggested, and based on the result of this experiment, I agree with the authors that this does not need to be included in the manuscript at this time. Clearly, there is some interesting work that can be done in the future to probe membrane contact sites involved in this process further, but that such experiments are beyond the scope of the current study. I have no further comments. This is a very exciting study that will be of significant and broad interest to cell biologists.